# A Deep Gravity model for mobility flows generation

Filippo Simini [1,2,3], Gianni Barlacchi[4], Massimilano Luca [5,6] & Luca Pappalardo [7✉]

The movements of individuals within and among cities influence critical aspects of our society, such as well-being, the spreading of epidemics, and the quality of the environment. When information about mobility flows is not available for a particular region of interest, we must rely on mathematical models to generate them. In this work, we propose Deep Gravity, an effective model to generate flow probabilities that exploits many features (e.g., land use, road network, transport, food, health facilities) extracted from voluntary geographic data, and uses deep neural networks to discover non-linear relationships between those features and mobility flows. Our experiments, conducted on mobility flows in England, Italy, and New York State, show that Deep Gravity achieves a significant increase in performance, especially in densely populated regions of interest, with respect to the classic gravity model and models that do not use deep neural networks or geographic data. Deep Gravity has good generalization capability, generating realistic flows also for geographic areas for which there is no data availability for training. Finally, we show how flows generated by Deep Gravity may be explained in terms of the geographic features and highlight crucial differences among the three considered countries interpreting the model's prediction with explainable AI techniques.

[1] University of Bristol, Department of Engineering Mathematics, Bristol, UK. [2] The Alan Turing Institute, London, UK. [3] Argonne Leadership Computing Facility, Argonne National Laboratory Lemont, Lemont, IL, USA. [4] Amazon Alexa, Berlin, Germany. [5] Fondazione Bruno Kessler, Trento, Italy. [6] Free University of Bolzano, Bolzano, Italy. [7] Institute of Information Science and Technologies (ISTI), National Research Council (CNR), Pisa, Italy. ✉email: luca.pappalardo@isti.cnr.it

Cities are complex and dynamic ecosystems that define where people live, how they move around, whom they interact with, and how they consume services[1–5]. Most of the world's population live now in urban areas, whose evolution in structure and size influences crucial aspects of our society such as the objective and subjective well-being[6–11] and the diffusion of innovations[4,12,13]. It is therefore not surprising that the study of human mobility has attracted particular interest in recent years[3,14–17], with a particular focus on the migration between cities and from rural to urban areas[18,19], the study and modeling of mobility patterns in urban environments[15,20–24], the estimation of city population[25–27], the migration induced by natural disasters, climate change, and conflicts[28–32], the prediction of traffic and crowd flows[14,33–37], and the forecasting of the spreading of epidemics[38–42]. Human mobility modelling has important applications in these research areas. Traffic congestion, domestic migration, and the spread of infectious diseases are processes in which the presence of mobility flows induces a net change of the spatial distribution of some quantity of interest (e.g., vehicles, population, pathogens). The ability to accurately describe the dynamics of these processes depends on our understanding of the characteristics of the underlying spatial flows and it is crucial to make cities and human settlements inclusive, safe, resilient, and sustainable[43–45].

Among all relevant problems in the study of human mobility, the generation of mobility flows, also known as flow generation[14,15,17], is particularly challenging. In simple words, this problem consists of generating the flows between a set of locations (e.g., how many people move from a location to another) given the demographics and geographic characteristics (e.g., population and distance) and without any historical information about the real flows.

Flow generation has attracted interest for a long time. Notably, in 1946 George K. Zipf proposed a model to estimate mobility flows, drawing an analogy with Newton's law of universal gravitation[46]. This model, known as the gravity model, is based on the assumption that the number of travelers between two locations (flow) increases with the locations' populations while decreases with the distance between them[15,47]. Given its ability to generate spatial flows and traffic demand between locations, the gravity model has been used in various contexts such as transport planning[48], spatial economics[18,49,50], and the modeling of epidemic spreading patterns[51–54].

Although the gravity model has the clear advantage of being interpretable by design and of requiring a few parameters, it suffers from several drawbacks, such as the inability to accurately capture the structure of the real flows and the greater variability of real flows than expected[15,47,55]. Since the gravity model relies on a restricted set of variables, usually just the population and the distance between locations, flows are generated without considering information that is essential to account for the complexity of the geographical landscape, such as land use, the diversity of points of interest (POIs), and the transportation network[56–59]. Therefore, we need more detailed input data and more flexible models to generate more realistic mobility flows. The former can be achieved by extracting a rich set of geographical features from data sources freely available online; the latter by using powerful nonlinear models like deep artificial neural networks. Deep-learning approaches exist for a different declination of the problem, namely flow prediction: they use historical flows between geographic locations to forecast the future ones, but they are not able to generate flows between pairs of locations for which historical flows are not available[14,33–37,60–65]. To what extent deep learning can generate realistic flows without any knowledge about historical ones is barely explored in the literature[14]. Finally, since deep-learning models are not transparent, explainability is crucial to gain a deeper understanding of the patterns underlying mobility flows. We may achieve this goal using explainable AI techniques[66–69], which unveil the most important variables overall as well as explain single flows between locations on the basis of their geographic characteristics.

We design an approach to flow generation that considers a large set of variables extracted from OpenStreetMap[70,71], a public and voluntary geographic information system. These variables describe essential aspects of urban areas such as land use, road network features, transportation, food, health, education, and retail facilities. We use these geographical features to train a deep neural network, namely the Deep Gravity model, to estimate mobility flows between census areas in England, Italy, and New York State. We prefer neural networks over other machine learning models because they are the natural extension of the state-of-the-art model for flow generation, i.e., the singly constrained gravity model[15,47], which corresponds to a multinomial logistic regression that is formally equivalent to a linear neural network with one softmax layer. Our approach is based on a nonlinear variant of the multinomial logistic regression obtained by adding hidden layers, which introduce nonlinearities and build more complex representations of the input geographic features.

We find that Deep Gravity outperforms flow generation models that use shallow neural networks or that do not exploit complex geographic features, with a relative improvement in the realism with respect to the classic gravity model of up to 66% (Italy), 246% (England), and 1076% (New York State) in highly populated areas, where flows are harder to predict because of the high number of relevant locations. Deep Gravity also has a good generalization capability, making it applicable to areas that are geographically disjoint from those used for training the model. Finally, we show how to explain Deep Gravity's predictions on the basis of the collected geographic features. This allows us to observe that, while in Italy and New York State the nonlinear relationship between population and distance captured by the model provides the strongest contribution to predict the flow probability, in England the interplay between the various geographic features plays a key role in boosting the model's predictions.

## Results

We define a geographical region of interest, $R$, as the portion of territory for which we are interested in generating the flows. Over the region of interest, a set of geographical polygons called tessellation, $\mathcal{T}$, is defined with the following properties: (1) the tessellation contains a finite number of polygons, $l_i$, called locations, $\mathcal{T} = \{l_i : i = 1, ..., n\}$; (2) the locations are nonoverlapping, $l_i \cap l_j = \emptyset$, $\forall i \neq j$; (3) the union of all locations completely covers the region of interest, $\bigcup_{i=1}^{n} l_i = R$.

The flow, $y(l_i, l_j)$, between locations $l_i$ and $l_j$ denotes the total number of people moving for any reason from location $l_i$ to location $l_j$ per unit time. As a concrete example, if the region of interest is England and we consider commuting (i.e., home to work) trips between England postcodes, a flow $y$(SW1W0NY, PO167GZ) may be the total number of people that commute every day between location (postcode) SW1W0NY and location PO167GZ. The total outflow, $O_i$, from location $l_i$ is the total number of trips per unit time originating from location $l_i$, i.e., $O_i = \sum_j y(l_i, l_j)$.

Given a tessellation, $\mathcal{T}$, over a region of interest $R$, and the total outflows from all locations in $\mathcal{T}$, we aim to estimate the flows, $y$, between any two locations in $\mathcal{T}$. Note that this problem definition does not allow to use flows within the region of interest as input data. That is, we cannot use a subset of the flows between the

locations in the region of interest neither historical information to generate other flows in the same region. This means that a model tested to predict flows in region $R$ must have been trained on a different region $R'$, nonoverlapping with $R$.

The most common metric used to evaluate the performance of flow generation models is the Sørensen-Dice index, also called Common Part of Commuters (CPC)[14,15,47], which is a well-established measure to compute the similarity between real flows, $y^r$, and generated flows, $y^g$:

$$CPC = \frac{2\sum_{i,j} min(y^g(l_i, l_j), y^r(l_i, l_j))}{\sum_{i,j} y^g(l_i, l_j) + \sum_{i,j} y^r(l_i, l_j)} \quad (1)$$

CPC is always positive and contained in the closed interval [0, 1] with 1 indicating a perfect match between the generated flows and the ground truth and 0 highlighting bad performance with no overlap. Note that when the generated total outflow is equal to the real total outflow, as for all the models we consider in this paper, CPC is equivalent to the accuracy, i.e., the fraction of trips' destinations correctly predicted by the model. In fact, when the generated total outflow is equal to the real total outflow, the denominator becomes $2\sum_{i,j} y^r(l_i, l_j)$ and the CPC measures the fraction of all trips that were assigned to the correct destination, i.e., the fraction of correct predictions or accuracy.

For a more comprehensive evaluation, we also use the Pearson correlation coefficient, the Normalized Root Mean Squared Error (NRMSE) and the Jensen-Shannon divergence (JSD), which measure the linear correlation, the error, and the dissimilarity between the distributions of the real and the generated flows, respectively[14] (see Supplementary Note 1 for details).

**Derivation of Deep Gravity**. Deep Gravity originates from the observation that the state-of-the-art model of flow generation, the gravity model[15,46,47,72], is equivalent to a shallow linear neural network. Based on this equivalence, we naturally define Deep Gravity by adding nonlinearity and hidden layers to the gravity model, as well as considering additional geographical features.

The singly constrained gravity model[15,47] prescribes that the expected flow, $\bar{y}$, between an origin location $l_i$ and a destination location $l_j$ is generated according to the following equation:

$$\bar{y}(l_i, l_j) = O_i p_{ij} = O_i \frac{m_j^{\beta_1} f(r_{ij})}{\sum_k m_k^{\beta_1} f(r_{ik})} \quad (2)$$

where $O_i$ is the origin's total outflow, $m_j$ is the resident population of location $l_j$, $p_{ij}$ is the probability to observe a trip (unit flow) from location $l_i$ to location $l_j$, $\beta_1$ is a parameter and $f(r_{ij})$ is called deterrence function. Typically, the deterrence function $f(r_{ij})$ can be either an exponential, $f(r) = e^{\beta_2 r}$, or a power-law function, $f(r) = r^{\beta_2}$, where $\beta_2$ is another parameter. In these two cases, the gravity model can be formulated as a Generalized Linear Model with a multinomial distribution[73]. Thanks to the linearity of the model, the maximum likelihood's estimate of parameters $\beta_1$ and $\beta_2$ in Eq. (2) can be found efficiently, for example using Newton's method, maximizing the model's loglikelihood:

$$Log\ L(\beta|y) \propto \ln\left(\prod_{i,j} p_{ij}^{y(l_i,l_j)}\right) = \sum_{i,j} y(l_i, l_j) \ln \frac{m_j^{\beta_1} f(r_{ij})}{\sum_k m_k^{\beta_1} f(r_{ik})}$$
$$= \sum_{i,j} y(l_i, l_j) \ln \frac{e^{\beta \cdot x(l_i, l_j)}}{\sum_k e^{\beta \cdot x(l_i, l_k)}} \quad (3)$$

where $y$ is the matrix of observed flows, $\beta = [\beta_1, \beta_2]$ is the vector of parameters and the input feature vector is $x(l_i, l_j) = concat[x_j, r_{ij}]$ for the exponential deterrence function ($x(l_i, l_j) = concat[x_j, \ln r_{ij}]$ for the power-law deterrence function) with $x_j = \ln m_j$. Note that

the negative of loglikelihood in Eq. (3) is proportional to the cross-entropy loss, $H = -\sum_i \sum_j \frac{y(l_i,l_j)}{O_i} \ln p_{i,j}$, of a shallow neural network with an input of dimension two and a single linear layer followed by a softmax layer.

This equivalence suggests to interpret the flow generation problem as a classification problem, where each observation (trip or unit flow from an origin location) should be assigned to the correct class (the actual location of destination) chosen among all possible classes (all locations in tessellation $\mathcal{T}$). In practice, for each possible destination in the tessellation, the model outputs the probability that an individual from a given origin would move to that destination. To compute the average flows from an origin, these probabilities are multiplied by the origin's total outflow. According to this interpretation, the gravity model is a linear classifier based on two explanatory variables, i.e., population and distance. The interpretation of the flow generation problem as a classification problem allows us to naturally extend the gravity model's shallow neural network introducing hidden layers and nonlinearities.

**Architecture of Deep Gravity**. To generate the flows from a given origin location (e.g., $l_i$), Deep Gravity uses a number of input features to compute the probability $p_{i,j}$ that any of the $n$ locations in the region of interest (e.g., $l_j$) is the destination of a trip from $l_i$. Specifically, the model output is a $n$-dimensional vector of probabilities $p_{i,j}$ for $j = 1, \ldots, n$. These probabilities are computed in three steps (see Fig. 1b).

First, the input vectors $x(l_i, l_j) = concat[x_i, x_j, r_{i,j}]$ for $j = 1, \ldots, n$ are obtained performing a concatenation of the following input features: $x_i$, the feature vector of the origin location $l_i$; $x_j$ the feature vector of the destination location $l_j$; and the distance between origin and destination $r_{i,j}$. For each origin location (e.g., $l_i$), $n$ input vectors $x(l_i, l_j)$ with $j = 1, \ldots, n$ are created, one for each location in the region of interest that could be a potential destination.

Second, the input vectors $x(l_i, l_j)$ are fed in parallel to the same feed-forward neural network. The network has 15 hidden layers of dimensions 256 (the bottom six layers) and 128 (the other layers) with LeakyReLu[74] activation function, $a$. Specifically, the output of hidden layer $h$ is given by the vector $z^{(0)}(l_i, l_j) = a(W^{(0)} \cdot x(l_i, l_j))$ for the first layer ($h = 0$) and $z^{(h)}(l_i, l_j) = a(W^{(h)} \cdot z^{(h-1)}(l_i, l_j))$ for $h > 0$, where $W$ are matrices whose entries are parameters learned during training.

The output of the last layer is a scalar $s(l_i, l_j) \in [-\infty, +\infty]$ called score: the higher the score for a pair of locations $(l_i, l_j)$, the higher the probability to observe a trip from $l_i$ to $l_j$ according to the model. Finally, scores are transformed into probabilities using a softmax function, $p_{i,j} = e^{s(l_i,l_j)}/\sum_k e^{s(l_i,l_k)}$, which transforms all scores into positive numbers that sum up to one. The generated flow between two locations is then obtained by multiplying the probability (i.e., the model's output) and the origin's total outflow.

The location feature vector $x_i$ provides a spatial representation of an area, and it contains features describing some properties of location $l_i$, e.g., the total length of residential roads or the number of restaurants therein. Its dimension, $d$, is equal to the total number of features considered. The location features we use include the population size of each location and geographical features extracted from OpenStreetMap[70,71] belonging to the following categories:

- Land-use areas (5 features): total area (in km²) for each possible land-use class, i.e., residential, commercial, industrial, retail, and natural;
- Road network (3 features): total length (in km) for each different types of roads, i.e., residential, main and other;

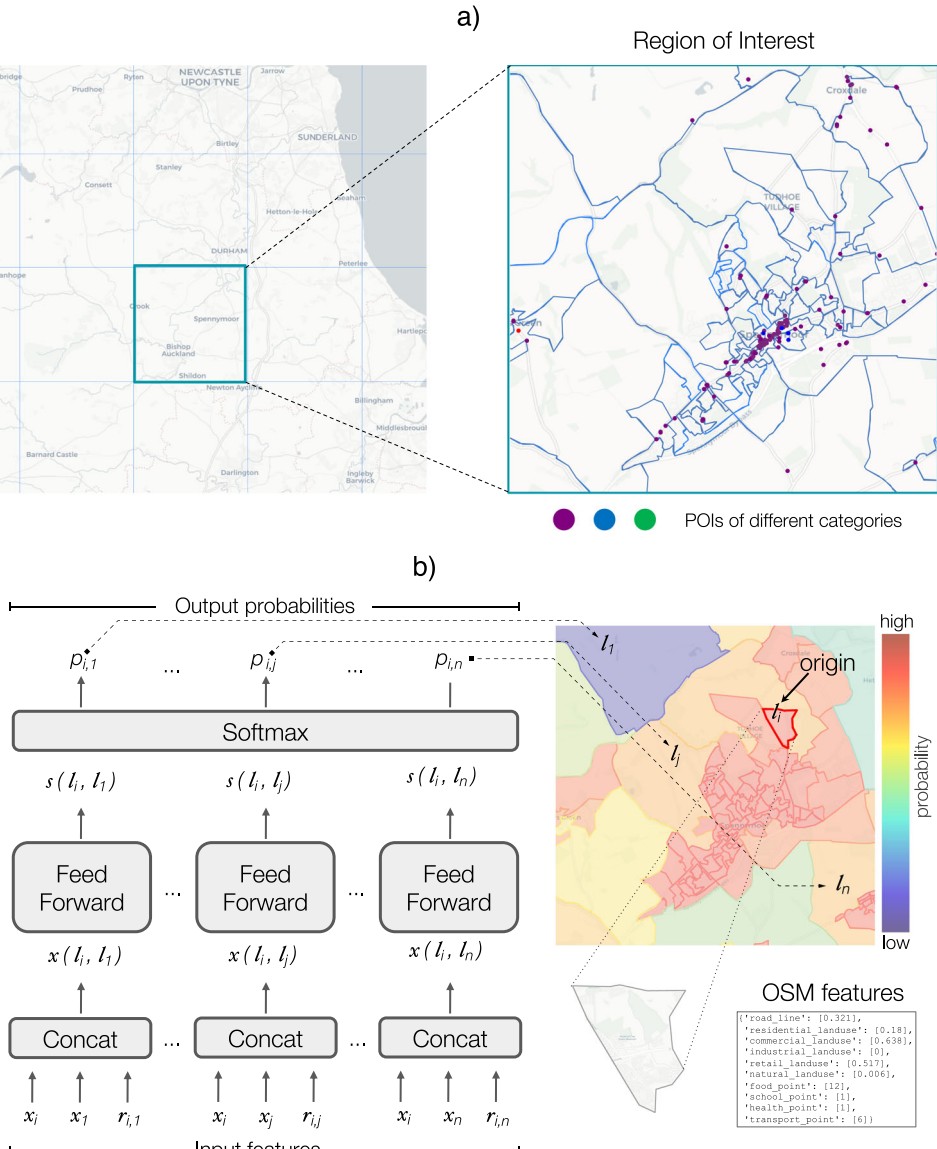

**Fig. 1 Flow generation and architecture of Deep Gravity. a** The geographic space is divided into regions of interest (squared tiles). Each region of interest is further split into locations using Output Areas (OAs, for England), Census Areas (CAs, for Italy), or Census Tracts (CTs, for New York State). Each location may have several Points Of Interest (POIs, the colored points), extracted from OpenStreetMap (OSM). During the training phase, half of the regions of interest are used for training the model and the remaining half to test the model's performance. **b** Architecture of Deep Gravity. The input features $x_i$ (feature vector of the origin location $l_i$), $x_j$ (feature vector of the destination location $l_j$), and $r_{i,j}$ (distance between origin and destination) are concatenated to obtain the input vectors $x(l_i, l_j)$. These vectors are fed, in parallel, to the same feed-forward neural network with 15 hidden layers with LeakyReLu activation functions. The output of the last hidden layer is a score $s(l_i, l_j) \in [-\infty, +\infty]$. The higher this score for a pair of locations $(l_i, l_j)$, the higher the probability to observe a trip from $l_i$ to $l_j$. Finally, a softmax function is used to transform the scores into probabilities $p_{i,j}$, which are positive numbers that sum up to one.

- Transport facilities (2 features): total count of Points Of Interest (POIs) and buildings related to each possible transport facility, e.g., bus/train station, bus stop, car parking;
- Food facilities (2 features): total count of POIs and buildings related to food facilities, e.g., bar, cafe, restaurant;
- Health facilities (2 features): total count of POIs and buildings related to health facilities, e.g., clinic, hospital, pharmacy;
- Education facilities (2 features): total count of POIs and buildings related to education facilities, e.g., school, college, kindergarten;

- Retail facilities (2 features): total count of POIs and buildings related to retail facilities, e.g., supermarket, department store, mall.

In addition, we include as feature of Deep Gravity the geographic distance, $r_{i,j}$, between two locations $l_i$ and $l_j$, which is defined as the distance measured along the surface of the earth between the centroids of the two polygons representing the locations. All values of features for a given location (excluding distance) are normalized dividing them by the location's area.

Each flow in Deep Gravity is hence described by 39 features (18 geographic features of the origin and 18 of the destination, distance between origin and destination, and their populations).

We also consider a light version of Deep Gravity in which we just count a location's total number of POIs without distinguishing among the categories (5 features per flow in total), and a heavy version of it in which we include the average of the geographic features of the $k$ nearest locations to a flow's origin and destination (e.g., 77 features per flow in total for $k = 2$). The performance of these two models is comparable to, or worse than, the performance of Deep Gravity (see Supplementary Note 2 and Supplementary Figs. 4 and 5).

The loss function of Deep Gravity is the cross-entropy:

$$H = -\sum_i \sum_j \frac{y(l_i, l_j)}{O_i} \ln p_{i,j}, \qquad (4)$$

where $y(l_i, l_j)/O_i$ is the fraction of observed flows from $l_i$ that go to $l_j$ and $p_{i,j}$ is the model's probability of a unit flow from $l_i$ to $l_j$. Note that the sum over $i$ of the cross-entropies of different origin locations follows from the assumption that flows from different locations are independent events, which allows us to apply the additive property of the cross-entropy for independent random variables. The network is trained for 20 epochs with the RMSprop optimizer with momentum 0.9 and learning rate $5 \cdot 10^{-6}$ using batches of size 64 origin locations. To reduce the training time, we use negative sampling and consider up to 512 randomly selected destinations for each origin location.

**Experiments**. We perform a series of experiments to estimate mobility flows in England (UK), Italy (EU), and New York State (US). In England and Italy, the mobility flows are among 885 and 1551 regions of interest, respectively, consisting of nonoverlapping square regions of 25 by 25 km², which cover the whole of the country. Half of these regions are used to train the models and the other half are used for testing. Each region of interest is further subdivided into locations: in England we use Output Areas (OAs) provided by the UK Census, in Italy we use Census Areas (CAs) provided by the Italian census. We also consider mobility flows among 5367 Census Tracts (CTs) provided by the United States Census Bureau in New York State extracted from millions of anonymous mobile phone users' visits to various places[75]. Supplementary Table 1 summarizes the characteristics of the datasets. For details on the definition of the regions of interest, the locations, their features, and the real flows used to train and validate the models, see Methods.

Our experiments aim to assess the effectiveness of the models in generating mobility flows within the region of interest belonging to the test set. Given the formal similarity between Deep Gravity (DG) and the gravity model (G), we use the latter as a baseline to assess Deep Gravity's improved predictive performance. Indeed, the gravity model is the state-of-the-art model for flow generation and it is thus preferred to a null model in which flows are evenly distributed at random across the edges of the mobility network[15,46,47]. Additionally, we define two hybrid models to understand the performance gain obtained by adding either multiple nonlinear hidden layers or complex geographical features to the gravity model:

- the Nonlinear Gravity model (NG) uses a feed-forward neural network with the same structure of Deep Gravity, but, similarly to the gravity model, its input features are only population and distance;
- the Multi-Feature Gravity model (MFG) has the same multiple input features of Deep Gravity, including various geographical variables extracted from OpenStreetMap but, similarly to the gravity model, these features are processed by a single-layer linear neural network;

Table 1 compares the performance of the models. For England, DG has CPC = 0.32, an improvement of 39% over MFG (CPC = 0.23), 166% over NG (CPC = 0.12), and 190% over G (CPC = 0.11) (see Table 1, Supplementary Fig. 1, and Supplementary Note 3). Note that DG's improvement on G is a common characteristic (see Fig. 3a–c). Although an overall CPC = 0.32 may seem low, we should consider that human mobility is a highly complex system: on the one hand, the number of factors influencing the decision underlying people's displacements are far more than those captured by the available features; on the other hand, mobility flows have an intrinsic random component and hence the prediction of a single event cannot be determined in a deterministic way. Figure 2a–c compares real flows with flows generated by DG and G on a region of interest in England. As suggested by the value of CPC computed on the flows in that region of interest, DG's network of flows is visually more similar to the real ones than G's one, both in terms of structure and distribution of flow values.

We obtain similar results for Italy and New York State (Table 1): DG performs significantly better than the other models, with an improvement in terms of global CPC over G of 66% (Italy) and 1076% (New York State). The improvement of DG over G is again spread in all areas of the two countries (Fig. 3d–i). This difference in the performance of DG among countries may be due to several factors, such as the differences in shapes and sizes of the spatial units, sparsity of flows, and mobility data sources.

To investigate the performance of the model in high and low populated regions, we split each country's regions of interest into ten equal-sized groups, i.e., deciles, based on their population, where decile 1 includes the regions of interest with the smaller population and decile 10 includes the regions of interest with the larger population, and we analyze the performance of the four models in each decile (Fig. 4 and Table 1). In England and Italy, all models degrade (i.e., CPC decreases) as the decile of the population increases, denoting that they are more accurate in sparsely populated regions of interest (Fig. 4a, c). This is not the case for New York State, in which the model's performance increases slightly as the decile of the population increases (Fig. 4e). Nevertheless, in all three countries, the relative improvement of DG with respect to G increases as the population increases (Fig. 4b, d, f). In other words, the performance of DG degrades less as population increases. This is a remarkable outcome because in highly populated regions of interest there are many relevant locations, and hence predicting the correct destinations of trips is harder. DG improves especially where current models are unrealistic.

The introduction of the geographic features (MFG) and of nonlinearity and hidden layers (DG) leads to a significant improvement of the overall performance. In England, the relative improvement of MFG and DG with respect to G is significant, with values of about 139% and 246%, respectively, in the last decile of population (see Fig. 4a,b and Table 1). Even in the first decile of the population, we find a relative improvement of DG with respect to G of 3%. Similarly, in the other two countries we have an improvement of MFG and DG over G of 14% and 66% (Italy) and of 351% and 1076% (New York State), respectively. Note that DG's improvement on G is a common characteristic, as DG improves on G in all the regions of interest for all countries (see Fig. 3 and 4). We find that the performances of NG and MFG are country specific. In England, MFG outperforms NG, as opposed to Italy and New York State where NG outperforms MFG (Fig. 4). Despite these country-specific differences, we observe a clear pattern in our results that is valid for all countries: G has always the worst performance and DG has always the best performance, while MFG and NG have intermediate performances. This confirms our hypothesis that the increased performance of DG originates from the interplay between a

**Table 1 Experimental results.**

| | | Decile of population | | | | | | | | | | Global metrics | | | |
|---|---|---|---|---|---|---|---|---|---|---|---|---|---|---|---|
| | | 1 | 2 | 3 | 4 | 5 | 6 | 7 | 8 | 9 | 10 | CPC | NRMSE | Corr. | JSD |
| **England** | | | | | | | | | | | | | | | |
| G | Mean CPC | 0.66 | 0.51 | 0.40 | 0.34 | 0.28 | 0.25 | 0.20 | 0.16 | 0.12 | 0.08 | 0.11 | 0.51 | 0.35 | 0.73 |
| | std CPC | 0.18 | 0.09 | 0.07 | 0.04 | 0.04 | 0.03 | 0.03 | 0.02 | 0.02 | 0.02 | | | | |
| NG | Mean CPC | 0.64 | 0.50 | 0.41 | 0.36 | 0.31 | 0.27 | 0.21 | 0.16 | 0.13 | 0.08 | 0.12 | 0.45 | 0.56 | 0.72 |
| | std CPC | 0.18 | 0.07 | 0.06 | 0.07 | 0.06 | 0.04 | 0.03 | 0.02 | 0.02 | 0.02 | | | | |
| | Rel. Imp. | -1.52 | -1.88 | 0.35 | 5.79 | 6.41 | 4.41 | 3.82 | 4.46 | 3.99 | 4.53 | | | | |
| MFG | Mean CPC | 0.66 | 0.52 | 0.45 | 0.41 | 0.36 | 0.36 | 0.32 | 0.30 | 0.26 | 0.19 | 0.23 | 0.47 | 0.48 | 0.65 |
| | std CPC | 0.17 | 0.09 | 0.07 | 0.05 | 0.04 | 0.04 | 0.06 | 0.05 | 0.04 | 0.04 | | | | |
| | Rel. Imp. | 1.29 | 1.55 | 13.46 | 20.11 | 26.89 | 43.01 | 61.43 | 87.83 | 105.64 | 139.46 | | | | |
| **DG** | Mean CPC | **0.67** | **0.57** | **0.50** | **0.48** | **0.44** | **0.45** | **0.41** | **0.39** | **0.35** | **0.28** | **0.32** | **0.41** | **0.61** | **0.60** |
| | std CPC | 0.17 | 0.07 | 0.06 | 0.06 | 0.04 | 0.05 | 0.05 | 0.05 | 0.04 | 0.05 | | | | |
| | Rel. Imp. | 3.20 | 11.72 | 24.91 | 41.47 | 54.35 | 75.76 | 108.47 | 143.54 | 174.97 | 246.88 | | | | |
| **Italy** | | | | | | | | | | | | | | | |
| G | Mean CPC | 0.26 | 0.38 | 0.41 | 0.37 | 0.31 | 0.29 | 0.27 | 0.24 | 0.21 | 0.13 | 0.18 | 0.48 | 0.49 | 0.69 |
| | std CPC | 0.27 | 0.14 | 0.09 | 0.09 | 0.08 | 0.07 | 0.06 | 0.06 | 0.05 | 0.05 | | | | |
| NG | Mean CPC | 0.31 | 0.44 | 0.48 | 0.43 | 0.38 | 0.35 | 0.34 | 0.30 | 0.25 | 0.15 | 0.21 | 0.45 | 0.57 | 0.67 |
| | std CPC | 0.31 | 0.14 | 0.10 | 0.10 | 0.08 | 0.07 | 0.06 | 0.07 | 0.06 | 0.06 | | | | |
| | Rel. Imp. | 19.30 | 14.63 | 16.41 | 16.82 | 19.76 | 22.26 | 23.67 | 22.93 | 19.93 | 19.45 | | | | |
| MFG | Mean CPC | 0.29 | 0.41 | 0.45 | 0.41 | 0.37 | 0.33 | 0.31 | 0.28 | 0.23 | 0.14 | 0.20 | 0.50 | 0.44 | 0.67 |
| | std CPC | 0.30 | 0.16 | 0.09 | 0.09 | 0.09 | 0.07 | 0.06 | 0.07 | 0.06 | 0.06 | | | | |
| | Rel. Imp. | 10.96 | 5.95 | 10.68 | 10.88 | 15.62 | 14.76 | 14.69 | 15.70 | 13.99 | 14.20 | | | | |
| **DG** | Mean CPC | **0.34** | **0.51** | **0.55** | **0.49** | **0.46** | **0.43** | **0.41** | **0.37** | **0.31** | **0.21** | **0.27** | **0.43** | **0.62** | **0.63** |
| | std CPC | 0.34 | 0.16 | 0.09 | 0.10 | 0.07 | 0.08 | 0.07 | 0.08 | 0.07 | 0.08 | | | | |
| | Rel. Imp. | 30.02 | 31.62 | 32.98 | 33.26 | 43.97 | 48.46 | 49.18 | 52.35 | 51.46 | 66.02 | | | | |
| **New York State** | | | | | | | | | | | | | | | |
| G | Mean CPC | 0.29 | 0.35 | 0.24 | 0.25 | 0.13 | 0.13 | 0.16 | 0.09 | 0.06 | 0.04 | 0.06 | 0.74 | 0.03 | 0.78 |
| | std CPC | 0.31 | 0.28 | 0.25 | 0.26 | 0.19 | 0.18 | 0.19 | 0.14 | 0.04 | 0.02 | | | | |
| NG | Mean CPC | 0.29 | 0.35 | 0.43 | 0.46 | 0.42 | 0.49 | 0.45 | 0.50 | 0.49 | 0.47 | 0.68 | 0.26 | 0.93 | 0.34 |
| | std CPC | 0.31 | 0.28 | 0.20 | 0.22 | 0.19 | 0.13 | 0.16 | 0.28 | 0.27 | 0.18 | | | | |
| | Rel. Imp. | 0 | 0 | 75.71 | 81.90 | 206.35 | 278.09 | 180.91 | 405.66 | 607.17 | 1031.14 | | | | |
| MFG | Mean CPC | 0.29 | 0.35 | 0.36 | 0.37 | 0.31 | 0.33 | 0.28 | 0.28 | 0.27 | 0.18 | 0.28 | 0.48 | 0.35 | 0.62 |
| | std CPC | 0.31 | 0.28 | 0.18 | 0.19 | 0.15 | 0.10 | 0.13 | 0.07 | 0.08 | 0.03 | | | | |
| | Rel. Imp. | 0 | 0 | 49.70 | 48.08 | 126.68 | 157.01 | 74.47 | 184.04 | 297.44 | 351.59 | | | | |
| **DG** | Mean CPC | **0.29** | **0.35** | **0.43** | **0.46** | **0.42** | **0.49** | **0.45** | **0.51** | **0.52** | **0.49** | **0.70** | **0.19** | **0.93** | **0.33** |
| | std CPC | 0.31 | 0.28 | 0.20 | 0.22 | 0.19 | 0.13 | 0.16 | 0.06 | 0.05 | 0.03 | | | | |
| | Rel. Imp. | 0 | 0 | 75.80 | 82.41 | 208.03 | 282.91 | 184.33 | 416.43 | 661.03 | 1076.93 | | | | |

Comparison of the performance, in terms of Common Part of Commuters (CPC), of Gravity (G), Nonlinear Gravity (NG), Multi-Feature Gravity (MFG), and Deep Gravity (DG), for England, Italy, and New York State, varying the decile of the population of the regions of interest (side size of 25 km). We also provide a global evaluation of the goodness of each model in terms of CPC, Pearson correlation coefficient, Normalized Root Mean Squared Error (NRMSE), and the Jensen-Shannon divergence (JSD) between the distribution of real and generated flows. For each model, and for each decile of population, we show the average CPC and the standard deviation of the CPC obtained over five runs of the model. For NG, MFG, and DG we also show the relative improvement in terms of CPC with respect to G. We put in bold the values over the deciles that correspond to the best mean CPC and relative improvement. Regardless of the evaluation metric, DG significantly improves on the other models in all the geographic areas considered.

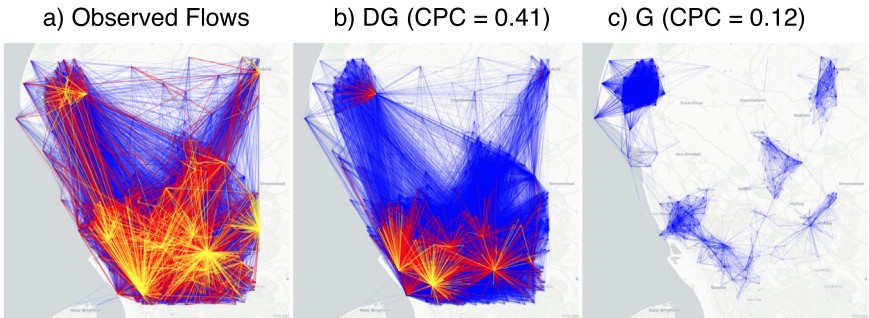

**Fig. 2 Real flows versus generated flows.** Visualization of the mobility network describing the observed flows (**a**), the flows generated by Deep Gravity (DG, panel **b**), and those generated by the gravity model (G, panel **c**) on a region of interest with 1001 locations (OAs) in the north of Liverpool, England, UK. Colored edges denote observed (**a**) or average flows (**b**, **c**): blue edges indicate flows with a number of commuters between 0 and 3, red edges between 3 and 5, and yellow edges above 5 commuters. CPC indicates the Common Part of Commuters. While both DG and G underestimate the flows, DG captures the overall structure of the flow network more accurately than G.

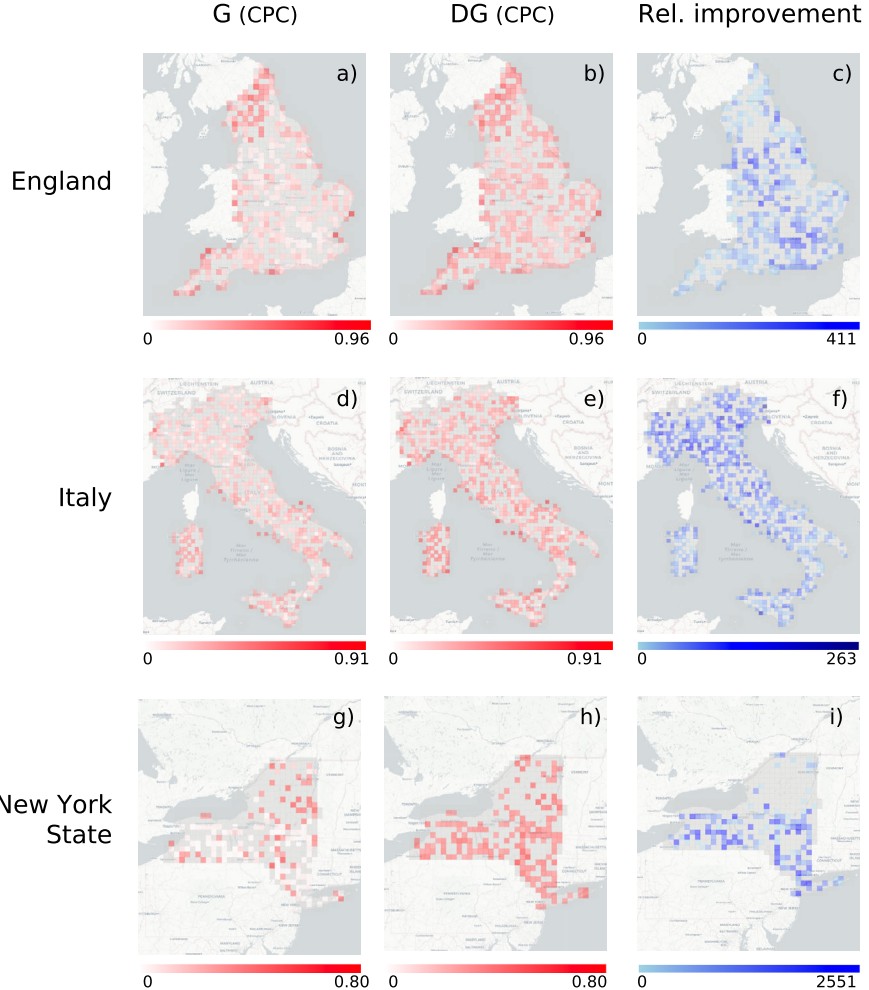

**Fig. 3 Performance of the models.** Common Part of Commuters (CPC) by region of interest (side size of 25 km) in England (**a**, **b**), Italy (**d**, **e**), and New York State (**g**, **h**) according to the gravity model (G) and Deep Gravity (DG). The CPC in each region of interest is the average CPC over the runs in which that region of interest has been selected in at least one test set. Gray regions of interest have never been selected in the test set. (**c**, **f**, **i**) Average relative improvement over five independent experiments in terms of CPC (in percentage) of DG with respect to G for each region of interest of side size 25 km in England, Italy, and New York State.

richer set of geographics features (present in MFG but not in NG) and the model's nonlinearity (present in NG but not in MFG).

The performance of all models does not change significantly if we use regions of interest of 10 by 10 km$^2$. In particular, all models have a $CPC_{25km}$ around 0.03 higher than $CPC_{10km}$ (see Supplementary Fig. 3, Supplementary Table 2, and Supplementary Note 3). However, the relative improvement of DG over G on the last decile is slightly smaller with a region of interest size of

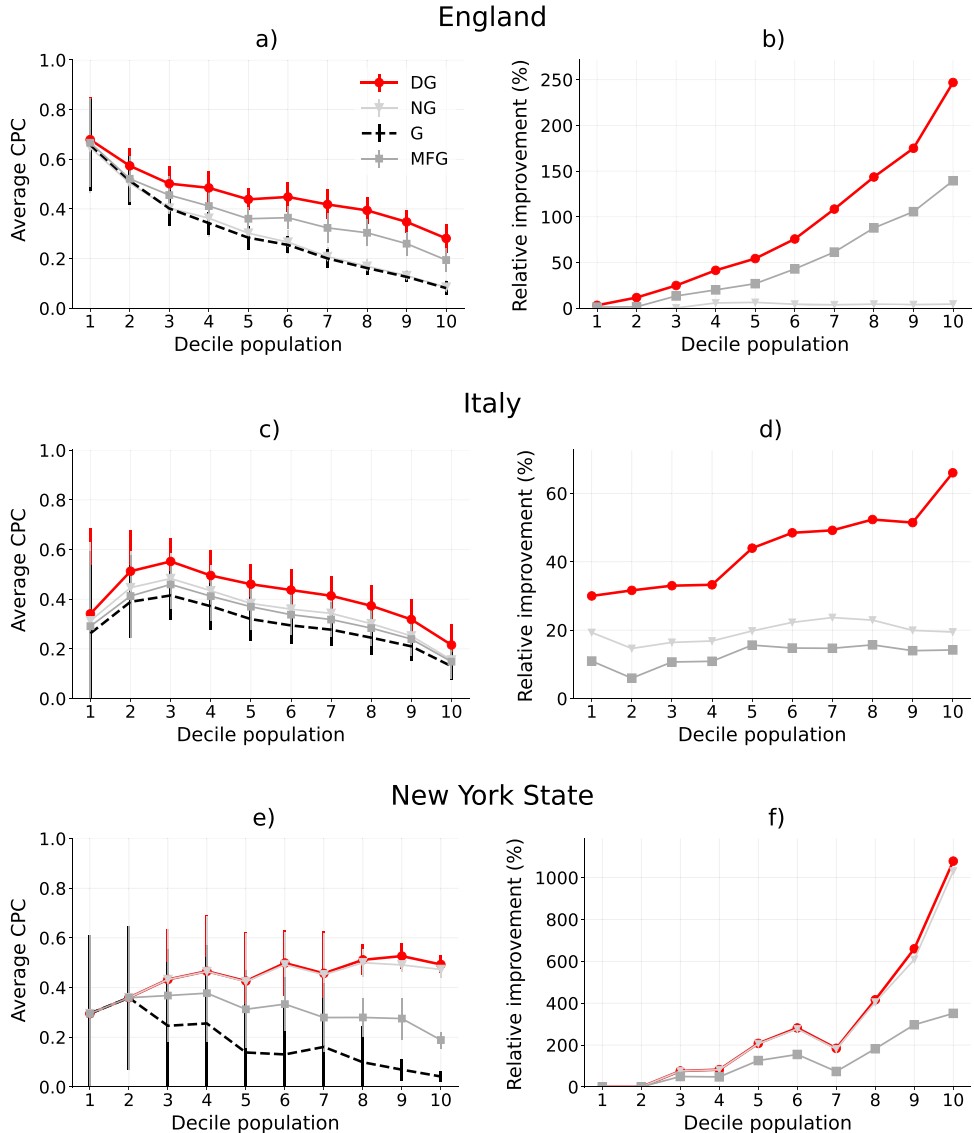

**Fig. 4 Performance of the models. a**, **c**, **e** Comparison of the performance in terms of Common Part of Commuters (CPC) of the gravity model (G, dashed line), Nonlinear Gravity (NG), Multi-Feature Gravity (MFG), and Deep Gravity (DG), varying the decile of the population and for regions of interest sizes of 25km for England (**a**), Italy (**c**) and New York State (**e**). Circles (DG), triangles (NG), and squares (MFG) indicate the average CPC for a decile. Error bars indicate the standard deviation of CPC of each decile. We run five independent experiments in which we train the models on 50% of randomly chosen tiles, stratifying according to the population in the decile, corresponding to 84,491.63 Output Areas for England, 200,746.15 Census Areas for Italy and 2118.93 Census Tracts for New York State. The remaining 50% of the tiles are used to evaluate the performance of the models in terms of CPC. DG is by far the approach with the best average CPC, regardless of the decile of the population. **b**, **d**, **f** Relative improvement with respect to G of NG, MFG, and DG, varying the decile of the population and for regions of interest of 25 by 25 km$^2$. The higher the population the higher is the improvement of DG with respect to G. For example, in England, for the last decile of the population, DG has an improvement of around 246% over G. Similarly, in Italy DG has a relative improvement of around 66%, while in New York State DG is around 1076% better than G in the last decile.

10km (Supplementary Table 2): for example, in England it is about 220%, i.e., about 26% less than the improvement on the same decile for a region of interest size of 25 km.

**Geographic transferability**. Neural networks trained on spatial data may suffer from low generalization capabilities when applied to different geographical regions than the ones used for training. In the previous experimental settings, it may happen that for a large city covered by multiple regions of interest (e.g., London, Manchester) some of its locations are used during the training phase, hence leading to a good performance when applied to test locations of the same city. To investigate the model generalization capability, we design specific training and testing datasets so that

a city is never seen during the training phase. This setting allows us to discover whether we can generate flows for a city where no flows have been used to train the model, a peculiarity that we cannot fully investigate if the model partially see a city (e.g., use some of the city flows during the training phase).

Given the nine England major cities, i.e., the so-called Core Cities[76] and London, the training dataset contains the locations and the information of eight cities and the test set contains information on the city excluded from the training. In particular, we select 15 regions of interest corresponding to London, eight to Leeds, seven to Sheffield, five to Birmingham, four to Bristol, Liverpool, Manchester and Newcastle, and three to Nottingham. In this way, we can test whether DG is able to generalize by

analyzing its performances according to a leave-one-city-out validation mechanism, i.e., generate flows on a city whose regions of interest never appear in the training set. We denote this implementation with Leave-one-city-out-DG (LDG).

LDG produces average CPCs that are remarkably close to the DG's ones (see Supplementary Fig. 2 and Supplementary Note 4). For instance, the average CPC slightly improves by testing LDG on London's locations using the locations of the other cities as training. We find similar results by testing the model on Newcastle, Liverpool and Nottingham. The average CPC slightly decreases when tested on Bristol and Sheffield, while it does not change significantly with respect to DG on Leeds, Birmingham and Manchester. The negligible difference between the performance DG and LDG shows that our model can generate flow probabilities also for geographic areas for which there is no data availability for training the model.

**Explaining generated flows.** Understanding why a model makes a certain prediction is crucial to interpret results, explain differences between models, and assess to what extent we understand the phenomenon under analysis[67–69]. Moreover, the Ethics Guidelines for Trustworthy AI of the EU High-Level Expert Group on AI suggest that the behavior of AI system should be transparent, explainable, and trustworthy[77,78].

We use SHapley Additive exPlanations (SHAP)[79,80] to understand how the input geographic features contribute to determine the output of Deep Gravity. SHAP is based on game theory[81] and estimates the contribution of each feature based on the optimal Shapley value[79], which denotes how the presence or absence of that feature change the model prediction of a particular instance compared to the average prediction for the dataset[66] (see Methods for details). We show some insights provided by SHAP for global explanations in the three countries considered (Fig. 5) and for local explanations for an origin-destination pair in England (Fig. 6).

From a global perspective (Fig. 5), one of the most relevant features with large Shapely values is the geographic distance: as expected, a large distance between origin and destination contributes to a reduction of flow probability, while a small distance leads to an increase. The population of the destination ("D: Population" in Fig. 5) is also globally relevant, especially in Italy and New York State. In England, however, in contrast with the usual assumption of the gravity model that the flow probability is an increasing function of the population, we find that population has a mixed effect, with high values of the population's feature (red points in Fig. 5a) that may also contribute to a decrease of the predicted flow. A possible explanation is that residential areas have a high population, but are not likely destinations of commuting trips, while other geographical features related to commercial and industrial land use, healthcare, and food are more relevant than population. For instance, locations having a large number of food facilities, retail, and industrial zones are predicted to attract commuters. On the other hand, locations with health-related POIs and commercial land use are predicted to have fewer commuters. Differently from England, in Italy and New York State (Fig. 5b,c) the populations in the origin and destination locations are the features with the strongest impact on the model output. In particular, both a small population in the origin and a large population in the destination increase the flow probability. The fact that populations and distance are more relevant than other geographic features in Italy and New York State explains why the Nonlinear Gravity model (NG) outperforms the Multi-Feature Gravity model (MFG) in these two countries: a deep-learning model that is able to capture the existing nonlinear relationship between populations and

distance can accurately predict the flow probabilities, while the other geographic features only bring a marginal contribution.

Finally, we show how to explain the contribution of each feature to Deep Gravity's prediction for a single origin-destination pair. We select two locations in England, E00137201 (population of 238 individuals) and E00137194 (population of 223 individuals) in a highly populated region of interest (in the 8th decile of population) situated in Corby, a city with about 50 thousands inhabitants (see Fig. 6a). We consider the two flows between them, i.e., from E00137201 to E00137194 and from E00137194 to E00137201. While the gravity model (G) generates identical flows because distances and populations are the same, Deep Gravity assigns different probabilities for the two flows and the Shapely values indicate that various geographical features (like transportation points and land use) are more relevant than population in this case (Fig. 6b,c). This example illustrates how DG's predictions for individual flows depend on the various geographical variables considered, and that the most relevant features for a specific origin-destination pair can differ from the most relevant features overall (Fig. 5a). Examples of local explanations for Italy and New York State are available in Supplementary Note 5.

## Discussion

The comparison of the performance of Deep Gravity with models that do not use nonlinearity or do not include the geographic information reveals several key results.

All models are generally more accurate on scarcely populated regions, which have fewer locations and trips' destinations are thus easier to predict. More importantly, in highly populated regions where there are many relevant locations and hence predicting the correct destinations of trips is harder, the improvement of Deep Gravity with respect to its competitors becomes much higher, suggesting that our model improves especially where current models fail. We observe that Deep Gravity still outperforms all its competitors even when using a smaller region of interest size.

The addition of the geographic features is crucial to boost the realism of our approach. In this regard, Deep Gravity's architecture allows for adding many other geographic features such as travel time with different transportation modalities, the structure of the underlying road network, or socio-economic information such as a neighborhood's house price, degree of gentrification, and segregation. Anyhow, our analysis clearly shows that it is the combination of deep neural networks and voluntary geographic information that significantly boosts the realism in the generation of mobility flows, paving the road to a new breed of data-driven flow generation models. In this regard, as depicted by the explanations extracted from DG, the impact of the voluntary geographic information and nonlinearity varies from country to country: while in England geographic information plays the strongest role in the model's performance, the nonlinearity predominates in Italy and New York State. More work is needed to delve into these interesting differences.

Deep Gravity is a geographic agnostic model able to generate flows between locations for any urban agglomeration, given the availability of appropriate information such as the tessellation, the total outflow per location, the population in the locations, and the information about POIs. This opens the doors to a set of intriguing questions regarding the model's geographical explainability and transferability[14]. Regarding explainability, while we use agnostic techniques to explain the role of geographic variables to the model's predictions, there is the need for more sophisticated explanations tailored for human mobility models. These explanations should take into account the peculiarities of flow

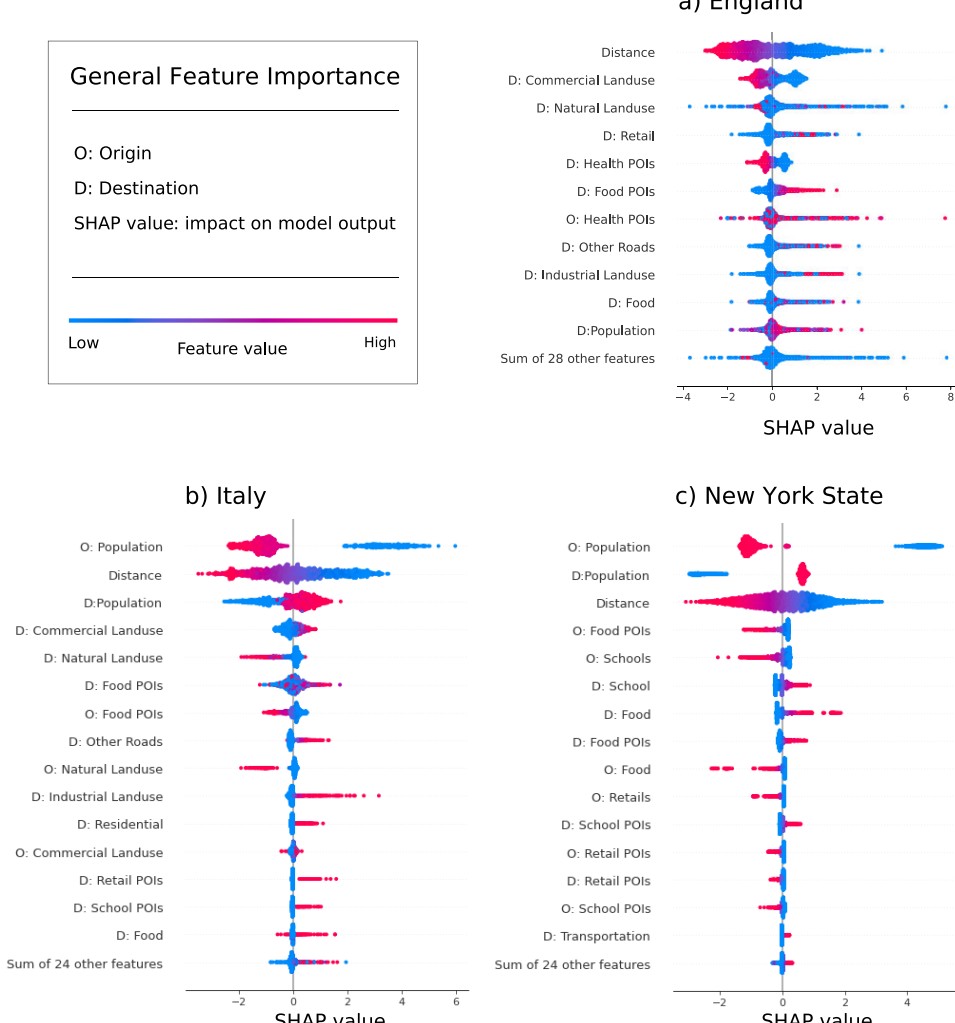

**Fig. 5 Explaining Deep Gravity.** Distribution of Shapely values for all features in Deep Gravity for England (**a**), Italy (**b**), and New York State (**c**). Features are reported on the vertical axis and are sorted from the most relevant on top to the least relevant on the bottom. Feature names starting with "D:" and "O:" indicate features of the destination and origin, respectively. Each point denotes an origin-destination pair, where blue points represent pairs where the feature has a low value and red points pairs with high values. For each variable, these points are randomly jittered along the vertical axis to make overlapping ones visible. The point's position on the horizontal axis represents the feature's Shapely value for that origin-destination pair, that is, whether the feature contributes to increase or decrease the flow probability for that pair. For example, high distances tend to be a travel deterrence while short distances are associated with an increment of commuters.

generation (e.g., spatiality, networking), providing more suitable global and local explanations of mobility flows. Regarding transferability, flow generation may be extended to include the generation of each location's total outflow to allow the application of the model to regions for which only the population and public POIs are available. Moreover, a future improvement of the model may consist in analyzing whether we can apply geographic transferability on other scales: Can we use rural areas flows to generate flows in cities? On the other hand, can we use cities' flows to generate flows in rural areas? And can we use a model trained on an entire country to generate flows on a different one?

## Methods

**Regions of interest.** First, we define a squared tessellation over the original polygonal shape of England. Formally, let $C$ be the polygon composed by $q$ vertices $v_1, \ldots, v_q$ that define the polygon boundary. We define the grid $G$ as the square tessellation covering $C$ with $L_x \times L_y$ regions of interest: $G = \{R_{ij}\}_{i=1,\ldots,L_x; j=1,\ldots,L_y}$,

where $R_{ij}$ is the square cell $(i, j)$ defined by two vertices representing the top-left and bottom-right coordinates. Depending on the nature of the problem, such tessellation can be formed with any triangle or quadrilateral tile or, as in the case of Voronoi tessellation, with tile defined as the set of points closest to one of the

points in a discrete set of defining points. In this paper, we build a square grid using the tessellation builder from the python library scikit-mobility[82], defining 885 regions of interest of 25 by 25 km², which cover the whole of England. Half of these regions are used to train the models and the other half are used for testing: the regions of interest have been randomly allocated to the train and test sets in a stratified fashion based on the regions' populations, so that the two sets have the same number of regions belonging to the various population deciles. Similarly, we have 1551 regions of interest of 25 by 25 km² covering Italy and 475 regions of 25 by 25 km² covering New York State.

**Locations.** The area covered by each region of interest is further divided into locations using a tessellation $\mathcal{T}$ provided by the UK Census in 2011. The UK Census defines 232,296 nonoverlapping polygons called census Output Areas (OAs), which cover the whole of England. By construction, OAs should all contain a similar number of households (125), hence in cities and urban areas where population density is higher, there is a larger number of OAs and they have a smaller size than average. For a given region of interest $R_{ij}$, its locations are defined as all OAs whose centroids are contained in $R_{ij}$. Unfortunately, information about real commuting flows at country level are provided by official statistics bureaus at the level of OAs only, which are administrative units of different shapes. It is not possible to aggregate or disaggregate flows between OAs onto flows between locations of the same size (e.g., using a squared tessellation) without introducing significant distortions in the data or obtaining aggregated locations of size much larger than the area of the largest OA. Regarding Italy, the Italian national statistics

## a) Localization of OAs E00137201 and E00137194

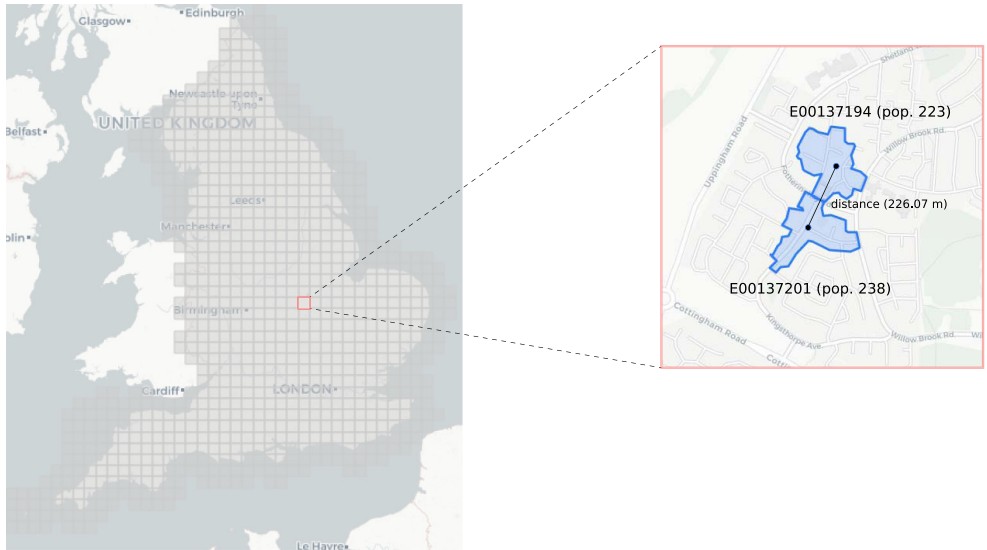

## b) Flow: E00137201 → E00137194    c) Flow: E00137194 → E00137201

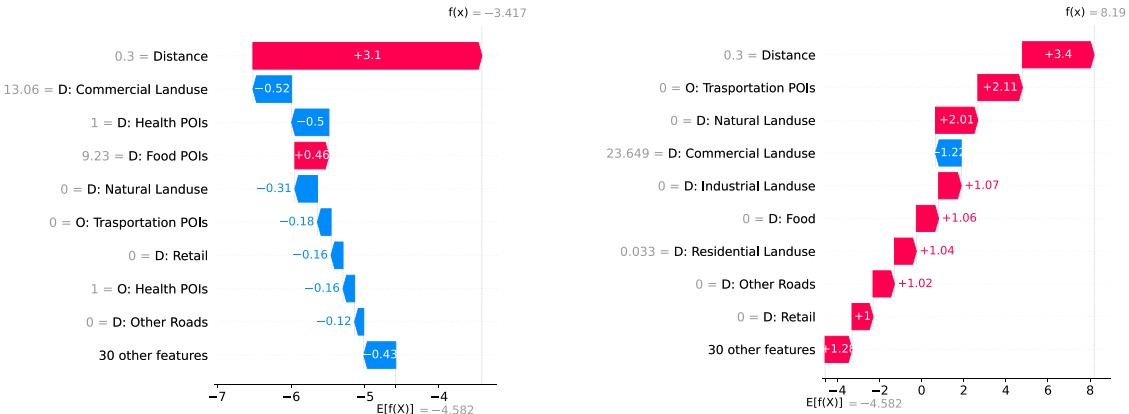

**Fig. 6 Explaining generated flows. a** Geographic position, shape, population, and distance between the Output Areas (OAs) E00137201 and E00137194, situated in Corby, England, a city with more than 50 thousand inhabitants (8th population decile). **b, c** Shapely values for the two flows between E00137201 (population of about 238 individuals) and E00137194 (population of about 223 individuals). Features are reported on the vertical axis, sorted from the most relevant on top to the least relevant on the bottom. The value of the feature is indicated in gray on the left of the feature name. The bars denote the contribution of each feature to the model's prediction (the number inside or near the bar is the Shapely value). The sum of the Shapely values of all features is equal to the model's prediction ($f(x)$ denotes the score): feature with positive (negative) Shapely values push the flow probability to higher (lower) values with respect to the model's average prediction $E[f(x)]$. Deep Gravity assigns different probabilities for the two flows and the Shapely values indicate that various geographical features are more relevant than the population in this case. The most relevant features for a specific origin-destination pair can differ from the most relevant features overall.

bureau (ISTAT) defines 402,678 nonoverlapping polygons known as Census Areas (CAs), which cover the entire country. Similarly to England, for a given region of interest, its locations are defined as all Census Areas (CAs) whose centroids are contained in $R_{i,j}$. Finally, the US Census Bureau defines Census Tracts (CTs) with characteristic similar to those of England and Italy. In particular, in New York State there are 5367 CTs.

**Location features**. We collect information about the geographic features of each location from OpenStreetMap (OSM)[70,71], an online collaborative project aimed to create an open source map of the world with geographic information collected from volunteers. The OSM data contain three types of geographical objects: nodes, lines and polygons. Nodes are geographic points, stored as latitude and longitude pairs, which represent points of interests (e.g., restaurants, schools). Lines are ordered lists of nodes, representing linear features such as streets or railways. Polygons are lines that form a closed loop enclosing an area and may represent, for example, land use or buildings. We use OpenStreetMap (OSM) data to compute the 18 geographic features for the origin and 18 for the destination. Regarding the

population, we include the number of inhabitants for each location as an input feature and we use the number of residents in each OA provided by the UK Census for the year 2011 and for each CA provided by the Italian Census for the year 2011, and the number of people estimated in each CT for New York State computed as the sum of outgoing flows from each CT.

**Mobility flows**. The UK Census collects information about commuting flows between OAs. We use the commuting flows collected by the UK Census in 2011 and consider flows that have origin location and destination location in the same region of interest only. The UK Census covers 30,008,634 commuters with an average flow of 1.78 and a standard deviation of 3.21. The Italian census covers 15,003,287 commuters with an average flow of 2.07 and a standard deviation of 4.27. Finally, in New York State, there are 41,070,279 commuters with an average of 66.86 people traveling between CTs and a standard deviation of 364.58.

**SHAP explanations**. SHAP (SHapley Additive exPlanations) applies a game theoretic approach to explain the output of any machine learning model[79]. It relies on the

Shapley values from game theory[81], which connect optimal credit allocation with local explanations. Shapley values consist in the average of the marginal contributions across all the permutations of the players solving a game. They are obtained by composing a combination of variables and their average change depending on the presence or absence of the variables to determine the importance of a single variable based on game theory[79]. Based on this idea, SHAP values are used as a unified measure of feature importance. The interpretation of the SHAP value for variable value $j$ is: the value of the $j$th variable contributed $\phi_j$ to the prediction of a particular instance compared to the average prediction for the dataset[66]. SHAP values allow us to give both a global and local explainability of Deep Gravity. In the first case, we use the collective SHAP values to understand which predictors (i.e., geographic variables, distance, populations) contributed either positively or negatively to the prediction. In the latter, we use single observations or smaller sets of observations (e.g., a specific decile) to both understand which features played a role in a specific prediction or, more in general, if the set of features used to predict flows in different deciles vary and how much it changes.

**Reporting summary**. Further information on research design is available in the Nature Research Reporting Summary linked to this article.

## Data availability

The commuting data for England are freely available at https://census.ukdataservice.ac.uk/use-data/guides/flow-data.aspx and https://census.ukdataservice.ac.uk/use-data/guides/boundary-data. he commuting data for Italy are freely available at http://dati open.istat.it/datasetPND.php and https://www.istat.it/it/archivio/104317. The flows data in New York State are freely available at github.com/GeoDS/COVID19USFlows and are described in Kang et al.[75].

## Code availability

The Python code of Deep Gravity is available at github.com/scikit-mobility/DeepGravity. The version of the code used to make the experiments in this paper is available at ref. [83].

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

## Acknowledgements

L.P. has been partially supported by EU project SoBigData++ grant agreement 871042. F.S. has been supported by EPSRC (EP/P012906/1). This research used resources of the Argonne Leadership Computing Facility, which is a DOE Office of Science User Facility supported under Contract DE-AC02-06CH11357. We acknowledge the OpenStreetMap contributors, OpenStreetMap data are available under the Open Database License and licensed as CC BY-SA https://creativecommons.org/licenses/by-sa/2.0/. We thank Daniele Fadda for his support on data visualization and plots design.

## Author contributions

F.S. designed the model and collected the data for England and Italy. M.L. collected the data for US. M.L. and L.P. performed the experiments. L.P. directed the study. All authors contributed to interpreting the results and writing the paper. G.B. developed this work prior joining Amazon.

## Competing interests

The authors declare no competing interests.
