## [Peer Review File · Nature Communications]

Reviewers' Comments:

Reviewer #1:

Remarks to the Author:

Summary:

This paper presents a neural network model for flow prediction. The presented model, Deep Gravity, takes rich features (extracted for both source and destination using openstreetMap), and train a multi-layer neural network to predict the outward flow distribution for each region. The authors have compared several methods, including the classic Gravity model as well as several variants of their own model and shown impressive improvements over those methods.

Strength:

- S1. The method is well motivated and sensible. Authors have analyzed the classic gravity model and shown its limitations by formulating it as a one-layer neural network. The proposed extensions of the gravity model, namely using multi-layer nonlinear neural networks and incorporating more features, are technically sensible and sound.
- S2. The empirical improvements over the baselines are quite impressive. With experiments on real data sets, the authors have shown their method can improve the baselines by up to more than 300% in terms of CPC. Such improvements are quite impressive.
- S3. The experiment results are detailed and convincing with both quantitative and qualitative analysis. In addition to showing overall improvements, the authors have also shown: a) the method has better generalization for those never-seen-before cities; 2) in which cases the proposed method is more superior to the baselines.
- S4. The paper is well written, the contents are organized in a logical way and flows smoothly. It is very easy to follow the key ideas and necessary details presented in the paper.

There are several concerns about this paper:

- D1. The key innovation and uniqueness of the proposed method is unclear. This model lacks discussion and comparisons to some related work. Applying deep neural networks for flow prediction is not new. Since 2016, multiple papers in the literature have been applying neural networks for flow prediction ([1] for example). The authors need to be more clear about their difference from those papers and why not compare with those existing methods.
- D2. Even though this model is an improvement over the classic revenue model. There are still several key limitations underlying the model's assumptions. First, the model still predicts for each region independently, without modeling the correlations between regions. But the correlations between regions are important to flow prediction. Second, the prediction is done in a static way, without modeling the dynamic flows among regions over time.

[1] Deep Spatio-Temporal Residual Networks for Citywide Crowd Flows Prediction, AAAI 2017

Reviewer #2:

Remarks to the Author:

The authors introduce a framework based on a deep neural network for modelling mobility network flows. The model takes as input a set of areas, the total number of outgoing travellers per area, the distances between any two areas, and the number and types of points-of-interest (e.g. restaurants/schools/transport facilities...) per area. As an output, the deep neural network model estimates the number of travellers between any two areas. The model presented is inspired by the standard 'gravity model' for mobility flows.

The authors used Census data collected in the UK to study the performance of the proposed framework against the gravity model. They show that the deep neural network model outperforms the gravity model, especially in densely populated areas.

The article is written very clearly, in a way that would be understandable by a relatively broad

audience. Analyses are performed thoroughly (see comments below for minor suggestions). The code and methodology provided with the paper will certainly help researchers and practitioners working with Human Mobility data to make better predictions. However, in the current state, the findings do not add much to what is already known about the mechanisms governing human mobility. It is not surprising that a model based on non-linear complex interactions between 20 features has better predictive power than a generalized linear model with two input variables. Additional analyses digging into the importance of features and their interactions could make this work more valuable.

Two main comments are the following:

Some sentences in the abstract and introduction seem to be misleading. In my understanding, the model allows to estimate mobility flows between areas, but only given that the number of outgoing trips per area is known. This information is, in general, hard to get. For this reason, sentences like: "When mobility data are not available for a particular region of interest, researchers must rely on mathematical models to generate mobility flows. Here we propose the Deep Gravity model as an effective method of flow generation..." need to be toned down. In my understanding, without mobility data (the total number of outgoing trips per location is mobility data), the model proposed in the paper can not estimate flows.

The framework is quite general, but only applied to one dataset. Results should be corroborated with different datasets. In particular, analyses based on locations with the same size would add value to the paper (see also comments below).

Below I provide detailed comments throughout the text. Line numbers were missing in the PDF, so I reported the sentences I am referring to in *italic*, followed by my comments (in the PDF attached).

- The flow, $y(i, j)$, between locations i and j denotes the number of trips (people moving) from location i to location j per unit time. For example, if our region of interest is in the England and our tessellation contains all UK postcodes, we can define the commuting flow $y(\text{SW1W0NY}, \text{PO167GZ})$ as the number of people that live in location (postcode) SW1W0NY and go to work in location PO167GZ every day.

Here the definition of trip is unclear. People moving from i to j per unit time may be commuters or may move for other reasons. Please clarify the definition of trip.

- This equivalence suggests to interpret the flow generation problem as a classification problem, where each observation (trip or unit flow from an origin location) should be assigned to the correct class (the actual location of destination) chosen among all possible classes (all locations in T).

What about the generation of the total outflow?

- The metric used to evaluate the performance of flow generation models is the Sørensen-Dice index, also called Common Part of Commuters (CPC) 13, 43

Here, since the total number of trips is an input of the model, isn't the denominator simply equal to $2 \cdot \sum_{(i,j)} y^{(i,j)}$? In this case (and in my understanding), this metric measures the fraction of all trips that were assigned to the correct destination. This would be something worth saying explicitly to help the reader. Especially in light of the analogy with a classification problem presented in the introduction, the authors could even replace the term CPC with 'accuracy', which is most widely understood.

- The location feature vector x_i provides a spatial representation of an area, and it contains features describing some properties of location l_i

Why isn't the population size an input feature (in analogy with the gravity model)?

- In addition, we included as a feature of Deep Gravity the distance, $r_{i,j}$, between two locations l_i, l_j , which is defined as the geographical distance (the distance measured along the surface of the earth) between the centroids of the two polygons representing the locations.

In the introduction, the authors talk about transportation as a key feature. Have you considered adding the time between origin and destination by different transportation modalities?

- All values of features for a given location (excluding distance) are normalized dividing them by the location's area.

Why do locations have different sizes? How different are they? A comparison with results obtained using locations with equal size would improve the paper.

- All models degrade (i.e., CPC decreases) as the decile of the population increases, denoting that they are more accurate in sparsely populated output areas.

May this be due to locations having different sizes? In sparsely populated areas, one has fewer locations and this makes the problem less complex. This could be better understood by comparing with a null model (e.g. even distribution of outgoing flows across edges).

- Table 1 compares the performance of the models. Overall, DG has $CPC = 0.31$.

It seems like the CPC measures accuracy. I find it interesting that despite the model being quite complex, the fraction of travellers assigned to the correct destination is relatively low. Can the authors comment on this?

- We do not find any significant improvement of NG with respect to G, despite the use of the non-linearity introduced by the deep neural network

Could the author comment on the fact that NG does not improve on G, but DG improves on MFG?

- This settings allows us to discover whether we can generate flows for a city for which we do not have any information, a peculiarity that we cannot fully investigate if the model partially "see" a city (e.g., use some of the locations of a city in the training phase).

If the total outgoing flow per location is an input of the model, the sentence above is misleading.

- The negligible difference between the performance DG and L-DG shows that our model can generate flows also for geographic areas for which no knowledge is available about the mobility flow patterns.

Same comment as above.

- In general, there is no single category which is significantly more important than the others.

What is the value gained by having different categories? What happens if one adds only the total number of POIs?

- Note that the addition of geographic features alone is not sufficient to explain the improvement of DG over the other models: the non-linearity plays a crucial role as well.

As per a previous comment, then how come NG does not perform better than G?

- Third, DG is a geographic agnostic model able to generate flows on urban agglomerations never seen before.

This claim should be toned down. The model helps estimating the flow network, given that the total outgoing flow per location is known. This is not a detail, because the total number of outgoing trips per location is not much easier to have than mobility network flows.

- Code Availability

Since this is also a methodological contribution, sharing the code with the paper would make this work more valuable. I can see that the instructions to access the code are provided with the paper, so maybe the section simply needs to be updated?

- Author contributions

Typo (FP)

- Figure 2

Why are there some grey cells in the map?

What are the yellow squares in panel (g)

- Figure 4

This is minor but the edges in panel 4 seem quite homogeneous despite the differences in the distances between locations,

Reviewer #3:

Remarks to the Author:

This paper introduces a new spatial interaction model to generate commuting flows based on a deep neural network architecture and voluntary geographic data. This model, called "Deep Gravity", thanks to its multiple non-linear hidden layers, is able to outperform the classical gravity model in the estimation of commuting flows in UK. The introduction of non-linear hidden layers seems to be particularly effective when considering additional geographical features. The paper is well-written and presents an interesting piece of work. However, I am not convinced that this work should be published in a high impact journal like Nature Communications.

My main problem is that this paper focuses only on the predictive power of the model without giving any insights regarding the underlying mechanisms of human mobility. It is not really surprising that if you add more complexity and inputs to a model you obtain better results. In my opinion, what it is really important when proposing a new model is to understand why this new model performs better at a global or a local scale. And how this knowledge could be used to generalize the approach. I did not find an answer to these questions in the manuscript. It would also be a good idea to add more case study sites from different countries and/or different data sources to rigorously assess the predictive power of the model. Finally, I would also recommend the authors to add a few goodness-of-fit metrics to compare the observed and the simulated network topologies (based for example on the number of link in common, the commuting distance distribution...).

Summary of Changes for Revision

Deep Gravity: enhancing mobility flows generation with deep neural networks and geographic information

Filippo Simini, Gianni Barlacchi, Massimiliano Luca, Luca Pappalardo

We thank the Referees for their insightful comments. In particular, we are glad that Referee #3 says: “analyses are performed thoroughly [...] the code and methodology provided with the paper will certainly help researchers and practitioners working with human mobility data to make better predictions”, and that Referee #2 says “the article is written very clearly, in a way that would be understandable by a relatively broad audience”. Finally, we are proud that Referee #1 says “the method is well motivated and sensible” and “the experiment results are detailed and convincing with both quantitative and qualitative analysis”.

We addressed all the concerns in the new version of the manuscript. Moreover, we replicated our experiments on other geographic places (Italy and New York State), and, motivated by a comment by Referee3, we added a new part to show how to explain the generated mobility flows from the geographic features. The new parts and those that have been modified with respect to the previous version are highlighted in blue.

We believe that the manuscript has improved significantly thanks to the comments of the reviewers, and we hope that now the paper meets the high standards to be published in Nature Communications.

Kind Regards, on behalf of all authors
Luca Pappalardo

Referee 1

We wish to thank Referee 1 for their comments, which gave us the opportunity to clarify important aspects of our work.

Point 1.1

The key innovation and uniqueness of the proposed method is unclear. This model lacks discussion and comparisons to some related work. Applying deep neural networks for flow prediction is not new. Since 2016, multiple papers in the literature have been applying neural networks for flow prediction ([1] for example). The authors need to be more clear about their difference from those papers and why not compare with those existing methods.

[1] Deep Spatio-Temporal Residual Networks for Citywide Crowd Flows Prediction, AAAI 2017

Response 1.1

We thank the Referee for this comment, which gives us the opportunity to highlight the uniqueness of our work.

We are aware that there is a strand of research dedicated to applying deep neural networks to human mobility (recently, we made a survey on this topic: <https://arxiv.org/abs/2012.02825>). In particular, the work mentioned by the Referee (Zhang et al. [1]) treats the problem of “crowd-flow prediction”, which differs from our problem (flow generation) in two important aspects:

1. Crowd-flow prediction aims at predicting future crowd flows, given the historical ones observed in the city. In contrast, flow generation aims to generate flows between locations *without* any knowledge about the historical flows between locations. In other words, crowd-flow prediction deals with a temporal aspect that is absent in flow generation.
2. Given a location, its crowd flow is the total inflow or outflow of that location, without any reference to the locations from which these people came. In contrast, in flow generation we are interested in directed flows, i.e., in generating flows between an origin and a destination.

While there is a vast literature on using the gravity model and its variants to solve flow generation, our approach was the only deep learning approach to flow generation at the time of submission.

We clarify the discussion above in the Introduction of the paper:

“Deep learning approaches exist for a different declination of the problem, namely flow prediction: they use historical flows between geographic locations to forecast the

future ones, but they are not able to generate flows between pairs of locations for which historical flows are not available. To what extent deep learning can generate realistic flows without any knowledge about historical ones is barely explored in the literature."

## Point 1.2

*Even though this model is an improvement over the classic revenue model. There are still several key limitations underlying the model's assumptions. First, the model still predicts for each region independently, without modeling the correlations between regions. But the correlations between regions are important to flow prediction. Second, the prediction is done in a static way, without modeling the dynamic flows among regions over time.*

## Response 1.2

We thank the Referee for these two important comments. We answer them in the following:

1. **On modelling the correlations between regions.**
   To investigate this aspect, we design a model (DG-Knn) that considers the features of the *k* nearest areas to a flow's origin and destination. The *k* nearest locations to the origin (destination) are computed by sorting all locations in decreasing order with respect to the geographic distance to the origin (destination) and then selecting the top k. We then train a model that exploits the geographic features of the origin, the geographic features of the destination, and the feature-wise average of the *k* nearest locations to the origin and the destination (DG-Knn is based on 77 features in total, while DG on 39 features).

   We perform experiments for *k* = 2, …, 9 and do not find any significant improvement of DG-Knn's performance with respect to DG, regardless of the value of *k*. In England, we observe an almost constant decrease of the relative improvement of DG-knn with respect to DG regardless of the *k*.

We replicate our experiments on two other datasets, describing flows between census tracts in Italy and the state of New York (see also Point 2.2 for a description of the new datasets). In Italy, we find a slight increase of the performances in the least populated areas and, in general, a deterioration of DG-knn up to the 10% with respect to DG. For the state of New York, DG-knn slightly improves on DG, but this improvement is in any case lower than 3%.

We describe in detail this experiment in Supplementary Information 2 for England and two additional datasets (Italy and the state of New York) and add the following sentence in section “Deep Gravity architecture” of the main manuscript:

“We also consider a more complex version of Deep Gravity, namely the DG-Knn model, which considers the includes the geographic features of the k nearest locations to a flow’s origin and destination (77 features in total), without finding any significant improvement with respect to the results presented in this paper (see Supplementary Information 3).”

2. On modelling the dynamic flows among regions over time.

The problem we address in the paper (flow generation) is a *generation* problem, and *not* a prediction problem. Indeed, the purpose of our model is to generate flows between pairs of regions for which *no historical data is available*. In other words, we aim to estimate the flow between two regions A and B for which we do not have any previous estimate of the flow between them. The absence of the historical observations impedes us to perform a predicting/forecasting task, and hence to model dynamic flows over time.

The problem the Referee presumably refers to is *flow prediction*, which differs from ours because it aims at predicting the value of future flows given historical observations about the same flows. In other words, given the historical flows between two regions A and B at times t_1, t_2, \dots, t_i , flow prediction aims at forecasting the value of the flow between A and B at time t_{i+1} .

We highlighted the difference between our problem (flow generation) and flow prediction in the section “Introduction” of the manuscript.

Referee 2

We thank Referee 2 for the careful reading of our paper and for their useful comments and suggestions, which we sincerely believe helped improve the paper significantly.

Point 2.1

Some sentences in the abstract and introduction seem to be misleading. In my understanding, the model allows to estimate mobility flows between areas, but only given that the number of outgoing trips per area is known. This information is, in general, hard to get. For this reason, sentences like: “When mobility data are not available for a particular region of interest, researchers must rely on mathematical models to generate mobility flows. Here we propose the Deep Gravity model as an effective method of flow generation...” need to be toned down. In my understanding, without mobility data (the total number of outgoing trips per location is mobility data), the model proposed in the paper can not estimate flows.

Response 2.1

We thank the Referee for this important point. Although the total number of outgoing trips per location is needed to obtain the quantitative estimation of the flows between locations, the flow generator may be used without that information, too. In this scenario, the model outputs the probabilities of going to each possible destination given the origin. The flow probabilities may be used in urban planning to perform what-if analysis and simulations by varying the total outflow per location.

To clarify this important aspect, we specified in the abstract that the model generates flow probabilities and rephrased the sentence highlighted by the Referee as follows (see also responses to points 2.4 e 2.16):

“Here we propose the Deep Gravity model as an effective method to generate flow probabilities that exploits many variables (e.g., land use, road network, transport, food, health facilities) extracted from voluntary geographic data and uses deep neural networks to discover non-linear relationships between those variables and mobility flows.”

Point 2.2

The framework is quite general, but only applied to one dataset. Results should be corroborated with different datasets. In particular, analyses based on locations with the same size would add value to the paper (see also comments below).

Response 2.2

We thank the Referee for this useful suggestion. We replicated the experiments on two new datasets of flows between census areas in Italy and the state of New York (US).

In Italy, the mobility flows are among 1551 regions of interest consisting of non-overlapping square regions of 25 by 25 km², which cover the whole of the country. Half of these regions are used to train the models and the other half are used for testing. Each region of interest is further subdivided into locations, according to the definition of census tracts provided by the Italian census bureau (ISTAT).

We also consider mobility flows among 5367 census tracts provided by the United States Census Bureau in the state of New York (US), extracted from millions of anonymous mobile phone users' visits to various places and aggregated at the census tract level [Kang2020]. We describe the characteristics of the datasets in Supplementary Table 1 and, in the section "Methods" of the manuscript, we provide details on the definition of the regions of interest, the Output Areas, the location features and the real flows used to train and validate the models on Italy and the state of New York.

[Kang2020] Kang, Y. et al., Multiscale dynamic human mobility flow dataset in the US during the COVID-19 epidemic. *Scientific data* 7, 1–13 (2020).

The results we obtain for Italy and New York are similar to those of England: DG performs significantly better than the other models, with an improvement in terms of global CPC over G of 50% in Italy and 1066% in the state of New York. Below, the Referee can find a table with the complete results for England, Italy, and the state of New York (it is Table 1 in the new version of the manuscript).

		Decile of Population										Global Metrics			
		1	2	3	4	5	6	7	8	9	10	CPC	NRMSE	Corr.	JSD
UK															
G	Mean CPC	0.66	0.51	0.40	0.34	0.28	0.25	0.20	0.16	0.12	0.08	0.11	0.51	0.35	0.73
	std CPC	0.18	0.09	0.07	0.04	0.04	0.03	0.03	0.02	0.02	0.02				
NG	Mean CPC	0.64	0.50	0.41	0.36	0.31	0.27	0.21	0.16	0.13	0.08	0.12	0.45	0.56	0.72
	std CPC	0.18	0.07	0.06	0.07	0.06	0.04	0.03	0.02	0.02	0.02				
	Rel. Imp.	-1.52	-1.88	0.35	5.79	6.41	4.41	3.82	4.46	3.99	4.53				
MFG	Mean CPC	0.66	0.52	0.45	0.41	0.36	0.36	0.32	0.30	0.26	0.19	0.23	0.47	0.48	0.65
	std CPC	0.17	0.09	0.07	0.05	0.04	0.04	0.06	0.05	0.04	0.04				
	Rel. Imp.	1.29	1.55	13.46	20.11	26.89	43.01	61.43	87.83	105.64	139.46				
DG	Mean CPC	0.67	0.57	0.50	0.48	0.44	0.45	41	0.39	0.35	0.28	0.32	0.41	0.61	0.60
	std CPC	0.17	0.07	0.06	0.06	0.04	0.05	0.05	0.05	0.04	0.05				
	Rel. Imp.	3.20	11.72	24.91	41.47	54.35	75.76	108.47	143.54	174.97	246.88				
Italy															
G	Mean CPC	0.26	0.38	0.41	0.37	0.31	0.29	0.27	0.24	0.21	0.13	0.18	0.48	0.49	0.69
	std CPC	0.27	0.14	0.09	0.09	0.08	0.07	0.06	0.06	0.05	0.05				
NG	Mean CPC	0.31	0.44	0.48	0.43	0.38	0.35	0.34	0.30	0.25	0.15	0.21	0.45	0.57	0.67
	std CPC	0.31	0.14	0.10	0.10	0.08	0.07	0.06	0.07	0.06	0.06				
	Rel. Imp.	19.30	14.63	16.41	16.82	19.76	22.26	23.67	22.93	19.93	19.45				
MFG	Mean CPC	0.29	0.41	0.45	0.41	0.37	0.33	0.31	0.28	0.23	0.14	0.20	0.50	0.44	0.67
	std CPC	0.30	0.16	0.09	0.09	0.09	0.07	0.06	0.07	0.06	0.06				
	Rel. Imp.	10.96	5.95	10.68	10.88	15.62	14.76	14.69	15.70	13.99	14.20				
DG	Mean CPC	0.34	51	0.55	0.49	0.46	0.43	0.41	0.37	0.31	0.21	0.27	0.43	0.62	0.63
	std CPC	0.34	0.16	0.09	0.10	0.07	0.08	0.07	0.08	0.07	0.08				
	Rel. Imp.	30.02	31.62	32.98	33.26	43.97	48.46	49.18	52.35	51.46	66.02				
State of New York															
G	Mean CPC	0.29	0.35	0.24	0.25	0.13	0.13	0.16	0.09	0.06	0.04	0.06	0.74	0.03	0.78
	std CPC	0.31	0.28	0.25	0.26	0.19	0.18	0.19	0.14	0.04	0.02				
NG	Mean CPC	0.29	0.35	0.43	0.46	0.42	0.49	0.45	0.50	0.49	0.47	0.68	0.26	0.93	0.34
	std CPC	0.31	0.28	0.20	0.22	0.19	0.13	0.16	0.05	0.02	0.03				
	Rel. Imp.	0	0	75.71	81.90	206.35	278.09	180.91	405.66	607.17	1031.14				
MFG	Mean CPC	0.29	0.35	0.36	0.37	0.31	0.33	0.28	0.28	0.27	0.18	0.28	0.48	0.35	0.62
	std CPC	0.31	0.28	0.18	0.19	0.15	0.10	0.13	0.07	0.08	0.03				
	Rel. Imp.	0	0	49.70	48.08	126.68	157.01	74.47	184.04	297.44	351.59				
DG	Mean CPC	0.29	0.35	0.43	0.46	0.42	0.49	0.45	0.51	0.52	0.49	0.70	0.19	0.93	0.33
	std CPC	0.31	0.28	0.20	0.22	0.19	0.13	0.16	0.06	0.05	0.03				
	Rel. Imp.	0	0	75.80	82.41	208.03	282.91	184.33	416.43	661.03	1076.93				

Similarly to what we did for England, to investigate the impact of population to the model performance, we split each country's regions of interest into ten equal-sized groups, i.e., deciles, based on their population, where decile 1 includes the regions of interest with the smaller population and decile 10 includes the regions of interest with the larger population, and we analyze the performance of the four models in each decile.

In Italy, all models degrade (i.e., CPC decreases) as the decile of the population increases, denoting that they are more accurate in sparsely populated output areas. This is not the case for the state of New York, in which the model's performance increases slightly as the decile of the population increases. Nevertheless, in all three countries, the relative improvement of DG with respect to G increases as the population increases. In other words, the performance of DG degrades less as population increases. Below, the Referee can find plots documenting the comparison between the performance of the models in the three datasets (it is Figure 3 in the new version of the manuscript).

Similarly to England, in the other two countries we have an improvement of MFG and DG over G of 14% and 66% (Italy) and of 351% and 1076% (New York State), respectively. DG's improvement on G is a common characteristic of all three datasets, as DG improves on G in all the tiles for all countries.

We added the above results and discussion in section "Experiments" of the new version of the manuscript.

Regarding the size of the locations, information about real commuting flows at country level are provided by official statistics bureaus at the level of output areas and census tracts only, which are administrative units of different shapes. It is not possible to aggregate or disaggregate flows between OAs onto flows between locations of the same size (e.g., using a squared tessellation) without introducing significant distortions in the data or obtaining aggregated locations of size much larger than the area of the largest OA. See also Point 2.8 for a detailed answer to this point.

Point 2.3

“The flow, $y(l_i, l_j)$, between locations l_i and l_j denotes the number of trips (people moving) from location l_i to location l_j per unit time. For example, if our region of interest is in the England and our tessellation contains all UK postcodes, we can define the commuting flow $y(SW1W0NY, PO167GZ)$ as the number of people that live in location (postcode) SW1W0NY and go to work in location PO167GZ every day.”

Here the definition of trip is unclear. People moving from i to j per unit time may be commuters or may move for other reasons. Please clarify the definition of trip.

Response 2.3

Thanks for this suggestion. In general, a flow between two locations indicates the total number of people moving between them for any reason (e.g., for commuting or tourism). Specifically, in our experiments in England and Italy, we use flows from the UK Census and Italian Census, which collect information about commuting flows between output areas and census tracts, respectively. For the state of New York, flows indicate movements of any kind between two US census tracts, extracted from mobile phone data.

We clarify the concept of flow in section “Results” as follows:

“The flow, $y(l_i, l_j)$, between locations l_i and l_j denotes the total number of people moving for any reason from location l_i to location l_j per unit time. As a concrete example, if the region of interest is the UK and we consider commuting (i.e. home to work) trips between UK postcodes, a flow $y(SW1W0NY, PO167GZ)$ may be the total number of people that commute every day between location (postcode) SW1W0NY and location PO167GZ.”

Moreover, we describe in detail the type of flow described by each dataset in subsections “Regions of interest” and “Locations” of section “Methods”.

Point 2.4

“This equivalence suggests to interpret the flow generation problem as a classification problem, where each observation (trip or unit flow from an origin location) should be

assigned to the correct class (the actual location of destination) chosen among all possible classes (all locations in T).”

What about the generation of the total outflow?

Response 2.4

The flow generator outputs the probabilities of moving from an origin to each possible destination. The origin's total outflow is used to transform the flow probabilities into average flows. To clarify this aspect, we modify the sentence above as follows:

“This equivalence suggests to interpret the flow generation problem as a classification problem, where each observation (trip or unit flow from an origin location) should be assigned to the correct class (the actual location of destination) chosen among all possible classes (all locations in T). In practice, for each possible destination in the tessellation, the model outputs the probability that an individual from a given origin would move to that destination location. To compute the average flows from an origin, these probabilities are multiplied by the origin's total outflow.”

As suggested by the Referee, the flow generation problem may be extended to include the generation of the total outflows to further enhance its geographical transferability. However, this is outside the scope of this paper and we leave it for future work.

We have added the following paragraph in the “Discussion” section discussing the possibility of including the generation of the total outflows:

“DG is a geographic agnostic model able to generate flows between locations for any urban agglomeration, given the availability of appropriate information such as the tessellation, the total outflow per location, the population in the output areas, and the information about POIs. This opens the doors to a set of intriguing questions regarding the limits of the model's geographical transferability. In this sense, the problem of flow generation may be extended to include the generation of each location's total outflow to allow the application of the model to regions for which only the population and public POIs are available.”

Point 2.5

The metric used to evaluate the performance of flow generation models is the Sørensen-Dice index, also called Common Part of Commuters (CPC) 13, 43

Here, since the total number of trips is an input of the model, isn't the denominator simply equal to $2 \cdot \sum_{(i,j)} (y^{i,j})$? In this case (and in my understanding), this metric measures the fraction of all trips that were assigned to the correct destination. This would be something worth saying explicitly to help the reader. Especially in light of the analogy with a

classification problem presented in the introduction, the authors could even replace the term CPC with 'accuracy', which is most widely understood.

Response 2.5

The Referee's reasoning is correct, and we agree that highlighting the equivalence between CPC and accuracy may improve the understanding of this measure and strengthen the analogy between flow generation and classification problems. We also note that the CPC can be used even when the generated total outflow is different from the real total outflow (like in globally constrained gravity models not considered in this paper) and hence cannot be interpreted as an accuracy.

To clarify this point we added the following sentence in the "Results" section:

"Note that when the generated total outflow is equal to the real total outflow, as for all the models we consider here, the CPC is equivalent to the accuracy, i.e., the fraction of trips' destinations correctly predicted by the model. In fact, when the generated total outflow is equal to the real total outflow, the denominator becomes $\sum_{i,j} y^{r(l_i, l_j)}$ and the CPC measures the fraction of all trips that were assigned to the correct destination, i.e., the fraction of correct predictions or accuracy. "

Point 2.6

"The location feature vector x_i provides a spatial representation of an area, and it contains features describing some properties of location l_i ".

Why isn't the population size an input feature (in analogy with the gravity model)?

Response 2.6

Thanks for this comment. We agree that population is an important feature and should be considered in the DG model as well. We remade all the experiments including the population feature and updated Table 1 of the manuscript accordingly.

With the inclusion of the population feature, we do not find any significant difference in the results with the respect to the previous version of the models. We clarified that we now use the population feature in the section "Architecture of Deep Gravity" of the manuscript:

"The location features we use include the population size of each location and geographical features extracted from OpenStreetMap."

Point 2.7

“In addition, we included as a feature of Deep Gravity the distance, r_{ij} , between two locations l_i, l_j , which is defined as the geographical distance (the distance measured along the surface of the earth) between the centroids of the two polygons representing the locations.”

In the introduction, the authors talk about transportation as a key feature. Have you considered adding the time between origin and destination by different transportation modalities?

Response 2.7

We thank the Referee for this interesting question. Travel time associated with different transportation modalities is surely an interesting feature to characterize the flow between two locations. We did not include this feature because it was not available from the census survey datasets we considered in our experiments. Given the flexibility of DG's architecture, any other feature, including travel times and costs with different transportation means, may be included to improve the characterization of locations and flows. We leave these improvements to our approach to future work.

We summarize the discussion above by adding the following paragraph in section “Discussion” of the manuscript:

“The addition of the geographic features is crucial to boost the realism of our approach. In this regard, DG's architecture allows for adding many other geographic features such as travel time with different transportation modalities, the structure of the underlying road network, or socio-economic information such as a neighborhood's house price, degree of gentrification, and segregation.”

Point 2.8

“All values of features for a given location (excluding distance) are normalized dividing them by the location's area.”

Why do locations have different sizes? How different are they? A comparison with results obtained using locations with equal size would improve the paper.

Response 2.8

Thank you for these questions, which give us the opportunity to clarify the definition of locations.

Actually, as Figure 1a of the manuscript shows for England (we report it below for the Referee's convenience), we first split the entire country into tiles of the same size (left part of

the figure). Then, each tile is further split into Output Areas (aka OAs, right part of the figure) of irregular shape, provided by the UK Census in 2011. By construction, OAs should all contain a similar number of households (125), hence in cities and urban areas where population density is higher, there is a larger number of Output Areas and they have a smaller size than average. Similar reasoning applies to the new datasets we add in the new version of the manuscript, corresponding to census tracts in Italy and New York State.

Unfortunately, information about real commuting flows at country level are provided by official statistics bureaus at the level of OAs or census tracts only, which are administrative units of different shapes. It is not possible to aggregate or disaggregate flows between OAs onto flows between locations of the same size (e.g., using a squared tessellation) without introducing significant distortions in the data or obtaining aggregated locations of size much larger than the area of the largest OA.

We clarified this point in the subsection “Locations” of section “Methods” of the manuscript.

Point 2.9

“All models degrade (i.e., CPC decreases) as the decile of the population increases, denoting that they are more accurate in sparsely populated output areas.”

May this be due to locations having different sizes? In sparsely populated areas, one has fewer locations and this makes the problem less complex. This could be better understood by comparing with a null model (e.g., even distribution of outgoing flows across edges).

Response 2.9

We thank the Referee for this useful comment. As we explain in response to Point 2.8, we first split the entire country into tiles of the same size and further split each tile into Output

Areas (OAs) of irregular shape, provided by the UK Census in 2011. Similarly we do for Italy and the state of New York.

As the Referee correctly points out, the most populated tiles are those with the highest number of OAs or census tracts (i.e., possible destinations of movements), making it more difficult for the model to determine the correct destination. We clarify this point in section “Results” of the manuscript:

“The relative improvement of DG with respect to G increases as the population increases. In other words, the performance of DG degrades less as population increases. This is a remarkable outcome because in highly populated regions of interest there are many relevant locations, and hence predicting the correct destinations of trips is harder.”

Regarding the comparison with a null model, we decided to compare DG with the gravity model of human mobility (G), which works significantly better than a null model in which flows are evenly distributed at random across the edges of the mobility network. We remark this discussion in section “Results” of the manuscript:

“Given the formal similarity between Deep Gravity (DG) and the Gravity model (G), we use the latter as a baseline to assess Deep Gravity’s improved predictive performance. Indeed, the Gravity model is the state-of-the-art model for flow generation and it is thus preferred to a null model in which flows are evenly distributed at random across the edges of the mobility network.”

Point 2.10

“Table 1 compares the performance of the models. Overall, DG has CPC = 0.31.”

It seems like the CPC measures accuracy. I find it interesting that despite the model being quite complex, the fraction of travellers assigned to the correct destination is relatively low. Can the authors comment on this?

Response 2.10

We thank the Referee for this interesting question. The Common Part of Commuters (CPC), also known as Sørensen-Dice index, is a well-established measure to compute the similarity between real flows and generated flows. It ranges between [0, 1], the higher the better, and it may be interpreted as the accuracy of the model, or more precisely as the realism of the generated flows.

In England, for DG we find that CPC=0.32. This value is significantly higher than that of G (CPC=0.11, with an improvement of 190% in the UK), which may be considered as a baseline for our model. Note that CPC varies with the population within the region of interest: it is higher in sparsely populated areas (e.g., 0.67 in the first decile for the UK) and lower in

densely populated areas (e.g., 0.28 in the last decile for the UK). We obtain similar results for Italy and the state of New York.

Although $CPC=0.32$ may seem low, one should consider that:

1. The CPC may vary with the composition and size of the locations: in densely populated areas such as London, for example, our results for G are in line with those in previous papers.
2. Human mobility is a highly complex system: on the one hand, the number of factors influencing the decision underlying people's displacements are far more than those captured by the available features; on the other hand, mobility flows have an intrinsic random component and hence the prediction of a single event cannot be determined in a deterministic way.

We summarized the discussion above in the section “Experiments” of the manuscript.

Point 2.11

“We do not find any significant improvement of NG with respect to G, despite the use of the non-linearity introduced by the deep neural network”

Could the author comment on the fact that NG does not improve on G, but DG improves on MFG?

Response 2.11

Thanks for this question, which gives us the opportunity to clarify our interpretation of this important aspect.

The analyses on two more countries, Italy and New York State, help us understand that the actual performances of NG and MFG are country specific. In England, we observe that MFG outperforms NG, as opposed to Italy and New York State where NG outperforms MFG. Despite these country-specific differences, we observe a clear pattern in our results that is valid for all countries: G has always the worst performance and DG has always the best performance, while MFG and NG have intermediate performances. This confirms our hypothesis that the increased performance of DG originates from the interplay between a richer set of geographics features (present in MFG but not in NG) and the model's nonlinearity (present in NG but not in MFG).

To clarify this point we added the following text in the “Experiments” section of the manuscript:

“We note that the performances of NG and MFG are country specific. In England, MFG outperforms NG, as opposed to Italy and New York State where NG outperforms MFG (Figure 3). Despite these country-specific differences, we observe a clear pattern in our results that is valid for all countries: G has always the worst

performance and DG has always the best performance, while MFG and NG have intermediate performances. This confirms our hypothesis that the increased performance of DG originates from the interplay between a richer set of geographics features (present in MFG but not in NG) and the model's nonlinearity (present in NG but not in MFG). ”

Point 2.12

“This settings allows us to discover whether we can generate flows for a city for which we do not have any information, a peculiarity that we cannot fully investigate if the model partially “see” a city (e.g., use some of the locations of a city in the training phase).”

If the total outgoing flow per location is an input of the model, the sentence above is misleading.

Response 2.12

The Referee is right, and we agree that the sentence above may be misleading. To avoid misunderstanding, we changed the sentence above as follows:

“This setting allows us to discover whether we can generate flows for a city where no flows have been used to train the model, a peculiarity that we cannot fully investigate if the model partially “see” a city (e.g., use some of the city flows during the training phase).”

Point 2.13

“The negligible difference between the performance DG and L-DG shows that our model can generate flows also for geographic areas for which no knowledge is available about the mobility flow patterns.”

Same comment as above.

Response 2.13

Again, the Referee is right and we agree that the sentence may lead to misunderstanding. We rephrased it as follows:

“The negligible difference between the performance DG and L-DG shows that our model can generate flow probabilities also for geographic areas for which there is no data availability for training the model.”

Point 2.14

*“In general, there is no single category which is significantly more important than the others.”
What is the value gained by having different categories? What happens if one adds only the total number of POIs?”*

Response 2.14

We thank the Referee for this useful suggestion. To investigate the contribution of different categories, we created a model (DG-sum) in which the only geographic feature is the total number of POIs in a location, regardless of the specific category they belong to. Each flow in DG-sum is hence described by 5 features only (sum of POIs of origin and destination, distance between origin and destination and their populations). We find that DG-sum has a lower performance than DG, regardless the decile of the population considered and the dataset (England, Italy, state of New York):

These results suggest that splitting POIs in categories brings a significant contribution, especially in England where DG has an improvement of about 141% in the last decile on DG-sum. While this improvement is less marked in Italy and the state of New York, we find an improvement of DG over DG-sum up to 17% (Italy) and 6% (state of New York).

We added the results above in Supplementary Information 3 and mentioned this experiment in section “Architecture of Deep Gravity” of the main manuscript:

“Each flow in Deep Gravity is hence described by 39 features (18 geographic features of the origin and 18 of the destination, distance between origin and destination, and their population). We also consider a light version of Deep Gravity, namely DG-sum, in which we just count a location’s total number of POIs without distinguishing among the categories (5 features per flow in total), and a more complex version of Deep Gravity, namely DG-Knn, which includes the geographic features of the k nearest locations to a flow’s origin and destination (77 features per flow in total). The performance of these two models is comparable to (DG-Knn) or worse than (DG-sum) the performance of Deep Gravity (see Supplementary Information 3).”

Point 2.15

“Note that the addition of geographic features alone is not sufficient to explain the improvement of DG over the other models: the non-linearity plays a crucial role as well.”

As per a previous comment, then how come NG does not perform better than G?

Response 2.15

Please see answer to Point 2.11.

Point 2.16

“Third, DG is a geographic agnostic model able to generate flows on urban agglomerations never seen before.”

This claim should be toned down. The model helps estimating the flow network, given that the total outgoing flow per location is known. This is not a detail, because the total number of outgoing trips per location is not much easier to have than mobility network flows.

Response 2.16

We thank the Referee for this comment: you are right and we agree that the sentence must be rephrased.

It is true that, for generating a region of interest’s flow network, the total outgoing flow as well as other information to run the model is needed (tessellation, population of output areas, POIs). We rephrased the sentence above in section Discussion as follows:

“Third, DG is a geographic agnostic model able to generate flows between locations for any urban agglomeration, given the availability of appropriate information such as the tessellation, the total outflow per location, the population in the output areas, and the information about POIs.”

Point 2.17

Code Availability

Since this is also a methodological contribution, sharing the code with the paper would make this work more valuable. I can see that the instructions to access the code are provided with the paper, so maybe the section simply needs to be updated?

Response 2.17

Thank you for this suggestion. We created a GitHub repository with the code of the models and the data to reproduce the experiments: <https://github.com/scikit-mobility/DeepGravity>.

Point 2.18

Author contributions

Typo (FP)

Response 2.18

Thanks for notifying us of this typo. Corrected.

Point 2.19

Figure 2

Why are there some grey cells in the map?

What are the yellow squares in panel (g)

Response 2.19

The two yellow squares in Figure 2g indicated regions of interest in England for which G works slightly better than DG. With the new experiments, in which we add the population size too as requested by the Referee, there no yellow cells anymore (see Figure 4 of the manuscript, that we report in the following):

Point 2.20

Figure 4

This is minor but the edges in panel 4 seem quite homogeneous despite the differences in the distances between locations

Response 2.20

Thanks for this comment. We remade the plots to highlight the differences between the origin networks of flows and those generated by G and DG better (see Figure below).

Referee 3

Point 3.1

My main problem is that this paper focuses only on the predictive power of the model without giving any insights regarding the underlying mechanisms of human mobility. It is not really surprising that if you add more complexity and inputs to a model you obtain better results. In my opinion, what is really important when proposing a new model is to understand why this new model performs better at a global or a local scale. And how this knowledge could be used to generalize the approach. I did not find an answer to these questions in the manuscript.

Response 3.1

The point raised by the Referee is very important and in the revised version we have made great efforts to address it. Understanding how to explain and interpret the predictions of complex models, like DG, is an active area of research and a wide variety of different methods have been recently proposed to address this issue.

Relying on explainability AI techniques, we now introduce and use Shapely values from game theory to quantify the contribution of each feature to the predictions of DG, at both global and local scales. This analysis is described in a new part in section “Experiments” and in the new Figure 5. Both are reported below:

“Understanding why a model makes a certain prediction is crucial to explain differences between models and interpret results. In this work, we use SHapley Additive exPlanations (SHAP) to understand how the input geographic features contribute to determine the output of Deep Gravity. SHAP is based on game theory and estimates the contribution of each feature based on the optimal Shapley value, which denotes how the presence or absence of that feature changes the model prediction of a particular instance compared to the average prediction for the dataset. In Figure 5, we show some insights provided by SHAP in England both for global (Figure 5a) and local (Figures 5b-d) explanations. From a global perspective, the most relevant feature (the one with largest Shapely value) is the geographic distance: as expected, a large distance between origin and destination contributes to a reduction of flow probability, while a small distance leads to an increase (Figure 5a). The population of the destination (“D: Population” in Figure 5) is also globally relevant. However, in contrast with the usual assumption of the gravity model that the flow probability is an increasing function of the population, we find that population has a mixed effect, with high values of the population feature (red points in Figure 5a) that may also contribute to a decrease of the predicted flow. This is probably because residential areas have a high population, but are not likely destinations of commuting trips. Instead, other geographical features related to commercial and industrial land use, healthcare, and food are more relevant than population. For

instance, locations having a large number of food facilities, retail, and industrial zones are predicted to attract commuters. On the other hand, locations with health-related POIs and commercial land use are predicted to have fewer commuters. Finally, we use Shapely values to explain the contribution of each feature to the model's prediction for a single origin-destination pair. Specifically, we select two locations in England, E00137201 and E00137194, with similar populations and in a highly populated region (in the 9th decile) and we consider the flows between them, from E00137201 to E00137194 and from E00137194 to E00137201. While the gravity model (G) generates identical flows because distances and populations are the same, Deep Gravity assigns different probabilities for the two flows and the Shapely values indicate that various geographical features (like transportation points and land use) are much more relevant than populations in this case (Figure 5b-d). This example illustrates how DG's predictions for individual flows depend on the various geographical variables considered and that the most relevant features for a specific origin-destination pair can be very different from the most relevant features overall (Figure 5a)."

Point 3.2

It would also be a good idea to add more case study sites from different countries and/or different data sources to rigorously assess the predictive power of the model.

Response 3.2

We thank the Referee for this useful suggestion. We replicated all the experiments on two new datasets, corresponding to flows between census areas in Italy and the state of New York (US).

In Italy, the mobility flows are among 1551 regions of interest consisting of non-overlapping square regions of 25 by 25 km², which cover the whole of the country. Half of these regions are used to train the models and the other half are used for testing. Similarly to the UK, each region of interest is further subdivided into locations, according to the definition of Output Areas (OAs) provided by the Italian census bureau (ISTAT).

We also consider mobility flows among 5367 census tracts provided by the United States Census Bureau in New York State (US), extracted from millions of anonymous mobile phone users' visits to various places and aggregated at the census tract level [Kang2020]. We describe the characteristics of the datasets in Supplementary Table 1 and we provide in the "Methods" section of the manuscript details on the definition of the regions of interest, the Output Areas, the location features and the real flows used to train and validate the models.

[Kang2020] Kang, Y. et al., Multiscale dynamic human mobility flow dataset in the US during the COVID-19 epidemic. *Scientific data* 7, 1–13 (2020).

The results we obtain for Italy and New York State are similar to those of England: DG performs significantly better than the other models, with an improvement in terms of global CPC over G of 50% in Italy and 1066% in New York State. Below, the Referee can find a table with the complete results for England, Italy, and New York State (it is Table 1 in the new version of the manuscript).

		Decile of Population										Global Metrics			
		1	2	3	4	5	6	7	8	9	10	CPC	NRMSE	Corr.	JSD
UK															
G	Mean CPC	0.66	0.51	0.40	0.34	0.28	0.25	0.20	0.16	0.12	0.08	0.11	0.51	0.35	0.73
	std CPC	0.18	0.09	0.07	0.04	0.04	0.03	0.03	0.02	0.02	0.02				
NG	Mean CPC	0.64	0.50	0.41	0.36	0.31	0.27	0.21	0.16	0.13	0.08	0.12	0.45	0.56	0.72
	std CPC	0.18	0.07	0.06	0.07	0.06	0.04	0.03	0.02	0.02	0.02				
	Rel. Imp.	-1.52	-1.88	0.35	5.79	6.41	4.41	3.82	4.46	3.99	4.53				
MFG	Mean CPC	0.66	0.52	0.45	0.41	0.36	0.36	0.32	0.30	0.26	0.19	0.23	0.47	0.48	0.65
	std CPC	0.17	0.09	0.07	0.05	0.04	0.04	0.06	0.05	0.04	0.04				
	Rel. Imp.	1.29	1.55	13.46	20.11	26.89	43.01	61.43	87.83	105.64	139.46				
DG	Mean CPC	0.67	0.57	0.50	0.48	0.44	0.45	41	0.39	0.35	0.28	0.32	0.41	0.61	0.60
	std CPC	0.17	0.07	0.06	0.06	0.04	0.05	0.05	0.05	0.04	0.05				
	Rel. Imp.	3.20	11.72	24.91	41.47	54.35	75.76	108.47	143.54	174.97	246.88				
Italy															
G	Mean CPC	0.26	0.38	0.41	0.37	0.31	0.29	0.27	0.24	0.21	0.13	0.18	0.48	0.49	0.69
	std CPC	0.27	0.14	0.09	0.09	0.08	0.07	0.06	0.06	0.05	0.05				
NG	Mean CPC	0.31	0.44	0.48	0.43	0.38	0.35	0.34	0.30	0.25	0.15	0.21	0.45	0.57	0.67
	std CPC	0.31	0.14	0.10	0.10	0.08	0.07	0.06	0.07	0.06	0.06				
	Rel. Imp.	19.30	14.63	16.41	16.82	19.76	22.26	23.67	22.93	19.93	19.45				
MFG	Mean CPC	0.29	0.41	0.45	0.41	0.37	0.33	0.31	0.28	0.23	0.14	0.20	0.50	0.44	0.67
	std CPC	0.30	0.16	0.09	0.09	0.09	0.07	0.06	0.07	0.06	0.06				
	Rel. Imp.	10.96	5.95	10.68	10.88	15.62	14.76	14.69	15.70	13.99	14.20				
DG	Mean CPC	0.34	51	0.55	0.49	0.46	0.43	0.41	0.37	0.31	0.21	0.27	0.43	0.62	0.63
	std CPC	0.34	0.16	0.09	0.10	0.07	0.08	0.07	0.08	0.07	0.08				
	Rel. Imp.	30.02	31.62	32.98	33.26	43.97	48.46	49.18	52.35	51.46	66.02				
State of New York															
G	Mean CPC	0.29	0.35	0.24	0.25	0.13	0.13	0.16	0.09	0.06	0.04	0.06	0.74	0.03	0.78
	std CPC	0.31	0.28	0.25	0.26	0.19	0.18	0.19	0.14	0.04	0.02				
NG	Mean CPC	0.29	0.35	0.43	0.46	0.42	0.49	0.45	0.50	0.49	0.47	0.68	0.26	0.93	0.34
	std CPC	0.31	0.28	0.20	0.22	0.19	0.13	0.16	0.05	0.02	0.03				
	Rel. Imp.	0	0	75.71	81.90	206.35	278.09	180.91	405.66	607.17	1031.14				
MFG	Mean CPC	0.29	0.35	0.36	0.37	0.31	0.33	0.28	0.28	0.27	0.18	0.28	0.48	0.35	0.62
	std CPC	0.31	0.28	0.18	0.19	0.15	0.10	0.13	0.07	0.08	0.03				
	Rel. Imp.	0	0	49.70	48.08	126.68	157.01	74.47	184.04	297.44	351.59				
DG	Mean CPC	0.29	0.35	0.43	0.46	0.42	0.49	0.45	0.51	0.52	0.49	0.70	0.19	0.93	0.33
	std CPC	0.31	0.28	0.20	0.22	0.19	0.13	0.16	0.06	0.05	0.03				
	Rel. Imp.	0	0	75.80	82.41	208.03	282.91	184.33	416.43	661.03	1076.93				

Similarly to what we did for England, to investigate the impact of population to the model performance, we split each country's regions of interest into ten equal-sized groups, i.e., deciles, based on their population, where decile1 includes the regions of interest with the smaller population and decile 10 includes the regions of interest with the larger population, and we analyze the performance of the four models in each decile.

In Italy, all models degrade (i.e., CPC decreases) as the decile of the population increases, denoting that they are more accurate in sparsely populated output areas. This is not the case for New York State, in which the model's performance increases slightly as the decile of the population increases. Nevertheless, in all three countries, the relative improvement of DG with respect to G increases as the population increases. In other words, the performance of DG degrades less as population increases. Below, the Referee can find plots documenting the comparison between the performance of the models in the three datasets (it is Figure 3 in the new version of the manuscript).

Similarly to England, in the other two countries we have an improvement of MFG and DG over G of 14% and 66% (Italy) and of 351% and 1076% (New York State), respectively. DG's improvement on G is a common characteristic of all three datasets, as DG improves on G in all the tiles for all countries.

We added the above results and discussion in the new version of the manuscript.

Regarding the size of the locations, information about real commuting flows at country level are provided by official statistics bureaus at the level of OAs only, which are administrative units of different shapes. It is not possible to aggregate or disaggregate flows between OAs

onto flows between locations of the same size (e.g., using a squared tessellation) without introducing significant distortions in the data or obtaining aggregated locations of size much larger than the area of the largest OA. See also Point 2.8 for a detailed answer to this point.

Point 3.3

I would also recommend the authors to add a few goodness-of-fit metrics to compare the observed and the simulated network topologies (based for example on the number of link in common, the commuting distance distribution...).

Response 3.3

We thank the Referee for this useful suggestion. Besides the CPC, which is by far the most common metric used to evaluate flow generation, we add the Pearson correlation coefficient, the Normalized Root Mean Squared Error (NRMSE), and a measure of distance (the Jensen-Shannon divergence or JSD) between the distribution of real flows and the distribution of generated flows. We added the formal definition and a description of these three metrics in Supplementary Information 1.

In the new version of the manuscript, we specify that we use these two additional evaluation metrics:

“For a more comprehensive evaluation, we also use the Pearson coefficient, the Normalized Root Mean Squared Error (NRMSE) and the Jensen-Shannon divergence (JSD), which measure the correlation, the error, and the distribution distance between the real and the generated flows, respectively (see Supplementary Information 4 for mathematical definitions).”

Reviewers' Comments:

Reviewer #1:

Remarks to the Author:

In this revision, the authors have addressed all my concerns in the previous review. I also appreciate that the authors have added experiments on extra data sets to showcase the generality of their method.

Reviewer #2:

Remarks to the Author:

I would like to congratulate the authors for the extensive revision, as they have done an impressive amount of work to address the reviewers' comments.

Given the extensive analyses aiming at testing the robustness of results across datasets, the soundness of the methodology and validity of the results are out of discussion.

Further, the authors have worked extensively on addressing the reviewers' comments regarding the significance of the study.

In the revised version, they have acknowledged that, as it is, the model can not be used without access to mobility data because it does not generate flows, and there is no guarantee of geographical transferability across countries. In my opinion, this implies that, as it is, the model has quite limited practical value, but I agree with the authors that this work paves the way for the development of future models based on deep learning that could be used in practice.

In terms of scientific understanding of Human Mobility, I find that the new section focusing on the interpretation of the model is one of the most interesting parts of the paper. For this reason, I would suggest the authors expand it, by showing the Shapley values for all the countries under study. I think the authors could also give more space to the results of this section in the abstract, introduction and discussion. It is interesting how amenities attract or repel commuters based on their type. And it would be interesting to compare the results obtained in England and Italy, for commuting patterns, against New York, where the mobility data is more detailed and comprehensive.

As a minor comment, it would be good to have a small description of SHAP in the methods or supplementary information. The authors should also explain why points spread along the y-axis in Figure 5a.

Reviewer #3:

Remarks to the Author:

The authors have addressed all my concerns.

Summary of Changes for Revision

Deep Gravity: enhancing mobility flows generation with deep neural networks and geographic information

Filippo Simini, Gianni Barlacchi, Massimiliano Luca, Luca Pappalardo

We thank the Referees for their insightful comments. In particular, we are proud that Referee #1 says: “ also appreciate that the authors have added experiments on extra data sets to showcase the generality of their method”, and that Referee #2 says “I would like to congratulate the authors for the extensive revision, as they have done an impressive amount of work to address the reviewers’ comments”.

We addressed all the concerns in the new version of the manuscript. In particular, we replicated our experiments about explanations on Italy and New York State, as requested by Referee #2. The new parts and those that have been modified with respect to the previous version are highlighted in blue.

Note that, since the previous title of the paper contained punctuation, we changed it to “A Deep Gravity model for mobility flows generation”.

We believe that the manuscript has improved significantly thanks to the comments of the reviewers, and we hope that now the paper meets the high standards to be published in Nature Communications.

Kind Regards, on behalf of all authors
Luca Pappalardo

Referee 1

“In this revision, the authors have addressed all my concerns in the previous review. I also appreciate that the authors have added experiments on extra data sets to showcase the generality of their method.”

We would like to thank Referee1 for their insightful comments and suggestions, and we are glad that they appreciated the new version of our paper.

Referee 2

Point 2.1

In the revised version, they have acknowledged that, as it is, the model can not be used without access to mobility data because it does not generate flows, and there is no guarantee of geographical transferability across countries. In my opinion, this implies that, as it is, the model has quite limited practical value, but I agree with the authors that this work paves the way for the development of future models based on deep learning that could be used in practice.

Response 2.1

Thanks for this comment. We highlight that the model may be improved into the direction desired by the Referee in the Discussion:

“Regarding transferability, flow generation may be extended to include the generation of each location's total outflow to allow the application of the model to regions for which only the population and public POIs are available.”

Point 2.2

In terms of scientific understanding of Human Mobility, I find that the new section focusing on the interpretation of the model is one of the most interesting parts of the paper. For this reason, I would suggest the authors expand it, by showing the Shapley values for all the countries under study. I think the authors could also give more space to the results of this section in the abstract, introduction and discussion. It is interesting how amenities attract or repel commuters based on their type. And it would be interesting to compare the results obtained in England and Italy, for commuting patterns, against New York, where the mobility data is more detailed and comprehensive.

Response 2.2

Thank you for this useful comment. As suggested by the Referee, we have expanded the discussion about explanations in the abstract, introduction, and discussion, and Supplementary Information 5. In particular, we added the following paragraph in the Abstract:

“Finally, we show how flows generated by Deep Gravity may be explained in terms of the geographic features and highlight crucial differences among the three considered countries interpreting the model's prediction with explainable AI techniques.”

and the following paragraphs in the Introduction:

“Finally, since deep learning models are not transparent, explainability is crucial to gain a deeper understanding of the patterns underlying mobility flows. We may achieve this goal using explainable AI techniques, which unveil the most important variables overall as well as explain single flows between locations on the basis of their geographic characteristics.

[...]

In particular, we observe that while in Italy and New York State the nonlinear relationship between population and distance captured by the model provides the strongest contribution to predict the flow probability, in England the interplay between the various geographic features plays a key role in boosting the model's predictions.”

In the Discussion, we added the following paragraphs:

“In this regard, as depicted by the explanations extracted from DG, the impact of the voluntary geographic information and non-linearity varies from country to country, as in England geographic information the former plays the strongest role in the model's performance while it predominates in Italy and New York. More work is needed to delve into this interesting dichotomy.

[...]

Regarding explainability, while we use agnostic techniques to explain the geographic variables' role to the model's predictions, there is the need for more sophisticated explanations tailored for human mobility models. These explanations should take into account the peculiarities of mobility tasks such as flow generation (e.g., spatiality, network), providing more suitable global and local explanations of mobility flows.”

Second, we have added the explanations for Italy and New York State in the new version of the manuscript.

“We use SHapley Additive exPlanations (SHAP) to understand how the input geographic features contribute to determine the output of Deep Gravity. SHAP is based on game theory and estimates the contribution of each feature based on the optimal Shapley value, which denotes how the presence or absence of that feature change the model prediction of a particular instance compared to the average prediction for the dataset (see Methods for details).

From a global perspective (Figure 5), one of the most relevant features with large Shapely values is the geographic distance: as expected, a large distance between origin and destination contributes to a reduction of flow probability, while a small distance leads to an increase. The population of the destination (D: Population' in Figure 5) is also globally relevant, especially in Italy and New York. In England however, in contrast with the usual assumption of the gravity model that the flow probability is an increasing function of the population, we find that population has a mixed effect, with high values of the population's features (red points in Figure 5a) that may also contribute to a decrease of the predicted flow. A possible explanation is that residential areas have a high population, but are not likely destinations of commuting trips, while other geographical features related to commercial and

industrial land use, healthcare, and food are more relevant than population. For instance, locations having a large number of food facilities, retail, and industrial zones are predicted to attract commuters. On the other hand, locations with health-related POIs and commercial land use are predicted to have fewer commuters. Differently from England, in Italy and New York (Figure 5b,c) the populations in the origin and destination locations are the features with the strongest impact on the model output. In particular, both a small population in the origin and a large population in the destination increase the flow probability. The fact that populations and distance are more relevant than other geographic features in Italy and NY explains why the Nonlinear Gravity model (NG) outperforms the Multi-Feature Gravity model (MFG) in these two countries: a deep-learning model that is able to capture the existing nonlinear relationship between populations and distance can accurately predict the flow probabilities, while the other geographic features only bring a marginal contribution.”

Point 2.3

As a minor comment, it would be good to have a small description of SHAP in the methods or supplementary information. The authors should also explain why points spread along the y-axis in Figure 5a.

Response 2.3

We thank the Referee for this useful suggestion. We added a description of SHAP in the Methods section (see below) and explained why points spread along the y-axis in Figure 5a.

“SHAP (SHapley Additive exPlanations) applies a game theoretic approach to explain the output of any machine learning model.

It relies on the Shapley values from game theory, which connect optimal credit allocation with local explanations. Shapley values consist in the average of the marginal contributions across all the permutations of the players solving a game. They are obtained by composing a combination of variables and their average change depending on the presence or absence of the variables to determine the importance of a single variable based on game theory. Based on this idea, SHAP values (SHapley Additive exPlanation values) are used in SHAP as a unified measure of feature importance. The interpretation of the SHAP value for variable value x_j is: the value of the j -th variable contributed ϕ_j to the prediction of a particular instance compared to the average prediction for the dataset.

SHAP values allow us to give both a global and local explainability of Deep Gravity. In the first case, we use the collective SHAP values to understand which predictors contributed (both positively or negatively) to the prediction. In the latter, we use single observations or smaller sets of observations (e.g., a specific decile) to both understand which features played a role in a specific prediction or, more in general, if the set of features used to predict flows in different deciles vary and how much it changes.”

Referee 3

"The authors have addressed all my concerns."

We thank the Referee for the overall appreciation of our work and for all their useful comments and suggestions, which helped us improve the paper.